# Inducible plasmid copy number control for synthetic biology in commonly used *E. coli* strains

Shivang Hina-Nilesh Joshi[1], Chentao Yong[1,2] & Andras Gyorgy [1]✉

The ability to externally control gene expression has been paradigm shifting for all areas of biological research, especially for synthetic biology. Such control typically occurs at the transcriptional and translational level, while technologies enabling control at the DNA copy level are limited by either (i) relying on a handful of plasmids with fixed and arbitrary copy numbers; or (ii) require multiple plasmids for replication control; or (iii) are restricted to specialized strains. To overcome these limitations, we present TULIP (TUnable Ligand Inducible Plasmid): a self-contained plasmid with inducible copy number control, designed for portability across various *Escherichia coli* strains commonly used for cloning, protein expression, and metabolic engineering. Using TULIP, we demonstrate through multiple application examples that flexible plasmid copy number control accelerates the design and optimization of gene circuits, enables efficient probing of metabolic burden, and facilitates the prototyping and recycling of modules in different genetic contexts.

Gene expression regulation has been instrumental in a wide variety of contexts, ranging from mitigating cytotoxicity to implementing cellular checkpoint control, and designing biomolecular feedback controllers[1-3]. At the transcriptional level, synthetic regulation has primarily been achieved through chemically inducible transcription factors (TFs) that have been repurposed from natural systems for metabolic engineering[4,5], protein production[6,7], and biosensing[8,9], thus offering powerful tools for systems and synthetic biology[10]. This repertoire is continuously expanded through genome mining approaches[11-13], directed evolution pipelines[10,14], and rational design strategies[15,16]. Complementing transcriptional control, a collection of powerful tools enables gene expression regulation at the translational level, including ligand-inducible riboswitches[17,18] and RNA-triggered regulators[19,20]. These techniques collectively enable precise and dynamic control of gene expression along the dimensions of transcriptional and translational activity.

Augmenting these tools, DNA plasmids represent a staple of bioengineering by providing a straightforward platform to introduce genetic constructs to cells, rendering them an essential tool for the plug-and-play design of gene circuits. However, unlike flexible control at the transcriptional and translational level, manipulating gene expression at the DNA level via plasmid copy number (PCN) is considerably more restricted. Typically, genetic circuits are inserted into a set of plasmids, each with a different fixed PCN, which are then individually introduced to the host organism, relying on time-consuming and error-prone cloning and transformation steps. To enable flexible PCN control, current solutions are either limited to specific strains[21-23], or require multiple plasmids[24-26], thus none of them offer (i) flexible and dynamic PCN regulation over time; (ii) portability across multiple strains; and (iii) self-containment for easy plug-and-play deployment of genetic circuits.

To overcome these limitations, here we present the DNA plasmid TULIP (TUnable Ligand Inducible Plasmid), equipped with the above mentioned three key properties. The design of TULIP is based on the auto-regulating mechanism of the pSC101 origin of replication, which is reconfigured and supplemented with additional regulatory components all harbored on TULIP to ensure self-containment and portability. Following fluorescence-based cell sorting to optimize key parameters involved in the regulation of the synthetic origin of replication, the PCN of TULIP can be induced upon the addition of Cuminic

[1]Division of Engineering, New York University Abu Dhabi, Abu Dhabi, United Arab Emirates. [2]Department of Chemical and Biomolecular Engineering, New York University, New York, NY, USA. ✉e-mail: andras.gyorgy@nyu.edu

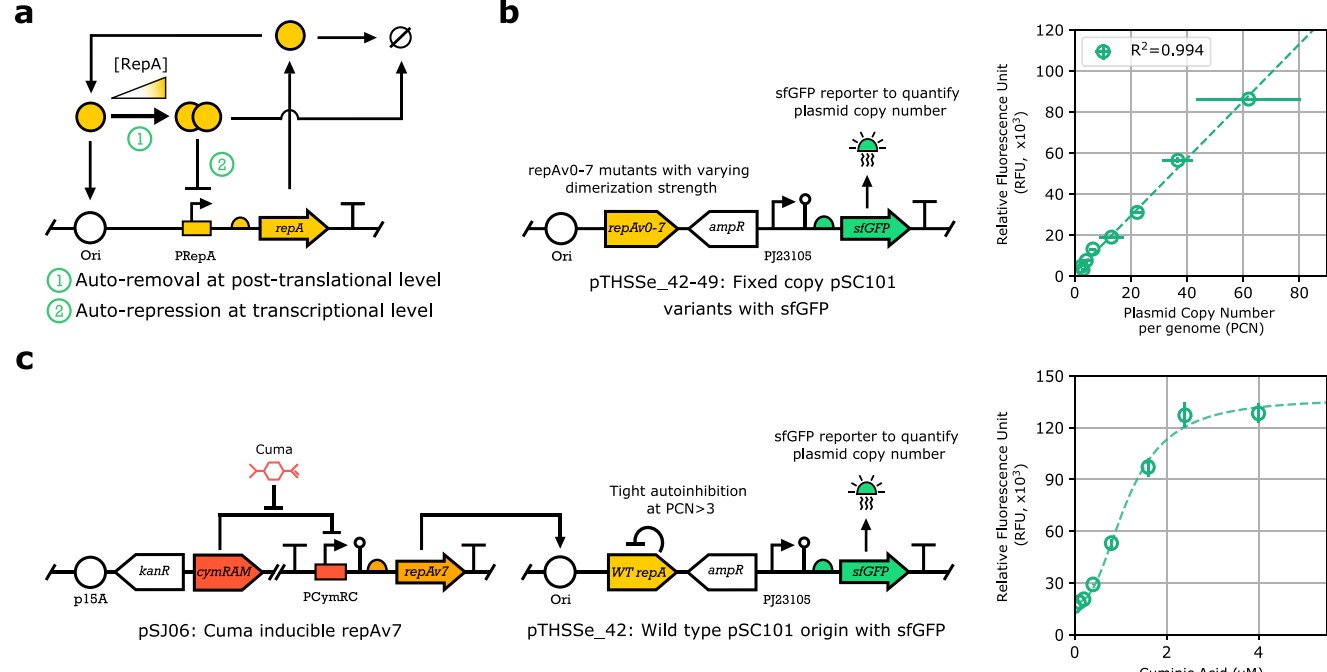

**Fig. 1 | PCN control offers a powerful avenue for regulating gene expression.** **a** Auto-regulating architecture of PCN control relying on the WT pSC101 origin of replication: RepA upregulates PCN in its monomeric form, and downregulates it as a homodimer. **b** Linear plasmid map of the pSC101-repA variants with fixed PCN (fPCN collection), each carrying identical constitutive sfGFP gene cassette and antibiotic resistance marker. PCN (measured via qPCR) and relative fluorescence intensity (measured via flow cytometry) for the fPCN collection show strong linear correlation. **c** Two-plasmid system with inducible PCN: the first plasmid harbors the Cuminic acid-inducible repAv7 gene, the second plasmid carries a sfGFP reporter (measured via flow cytometry) and a WT pSC101 origin, where the WT RepA exhibits tight auto-inhibition at PCNs greater than approximately 3[73]. By inducing the expression of RepAv7, a RepA mutant gene with severely diminished dimerization capacity, we can control the PCN of the second reporter plasmid.

acid (Cuma) over a broad dynamic range, spanning up to two orders of magnitude. TULIP displays robust behavior in commonly used *Escherichia coli* strains (NEBStable, DH10B, NEBExpress, BW25113, and MG1655) and growth media (M9-Glucose, M9-Glycerol, Lysogeny Broth, and Super Optimal Broth). As we illustrate through multiple application examples, TULIP enables dynamic PCN control not only to accelerate the design and optimization of gene circuits, but also to facilitate the prototyping and recycling of modules in different genetic contexts.

This paper is organized as follows. We start by presenting the regulatory architecture and the design and screening steps underpinning the implementation of TULIP on a single plasmid. Second, we demonstrate key properties of TULIP: portability, dynamic PCN control, and plasmid stability. Next, we illustrate its practical applicability and impact considering the design, optimization, re-use, and prototyping of common synthetic biology modules. Finally, we demonstrate how TULIP's modular architecture can be leveraged in combination with a CRISPRi-module to extend the input of PCN control beyond Cuma, increasing its versatility.

## Results
### Reconfiguring the pSC101 origin for inducible PCN control
To verify the potential of leveraging PCN to control gene expression, we first characterized constitutive sfGFP production using a library of plasmids (fPCN collection) with a range of fixed PCNs[24]. To this end, we considered the wild type (WT) pSC101 origin of replication, equipped with a stringent control mechanism ensuring that the plasmid is maintained at approximately 3 copies per cell[27,28], relying on the tight autoregulation mechanism of RepA, a plasmid replication initiation factor that acts at the Ori region (Fig. 1a). In this scheme, PCN is determined by the balance of two opposing forces. First, RepA in its monomeric form binds to the Ori site and promotes plasmid

replication initiation, thus upregulating PCN. Second, at higher expression levels, dimerized RepA acts as a transcriptional repressor inhibiting its own expression, in addition to sequestering its monomeric form, thus downregulating PCN by dilution.

While additional factors also affect the pSC101 copy regulation[29,30], PCN is largely determined by the ratio of monomeric and dimeric forms of RepA at the equilibrium[31]. Introducing mutations at the RepA dimerization interface (e.g., via amino acid substitution) alters the binding kinetics of the RepA monomers and modulates the dissociation constant, thus resulting in mutant variants equipped with different fixed PCNs[24,32,33]. For instance, in NEBStable *E. coli* cells the fPCN collection offers an approximately 20-fold PCN range with proportional gene expression levels of the reporter protein sfGFP (Fig. 1b) without noticeable perturbation to growth rate (Supplementary Fig. 1).

To demonstrate that PCN can be flexibly regulated, we next constructed a two-plasmid system (Fig. 1c). The first plasmid harbors the RepA coding sequence from Fig. 1b, expressed from the Cuminic acid-inducible promoter PCymRC. Regarding the repA variant on the first plasmid, we selected the mutant with the weakest dimerization strength (RepAv7) to effectively remove the negative feedback present in Fig. 1a. This plasmid was co-transformed with a WT pSC101 plasmid carrying a constitutive sfGFP gene, together with the WT repA gene, yielding negligible basal RepA expression due to tight negative feedback[27,28]. The architecture in Fig. 1c leverages the modular structure of the pSC101 origin: upon induction with Cuminic acid, RepAv7 is expressed from the first plasmid, then interacts as a monomer with the Ori site of the second plasmid, thus upregulating its PCN. This is confirmed in Fig. 1c, yielding an expression range comparable to the one observed when using the fPCN collection in Fig. 1b (the slight difference in the absence of Cuminic acid is most likely due to leaky expression of RepAv7 from PCymRC). Therefore, the intermediate construct in Fig. 1c demonstrates that inducible RepAv7 expression

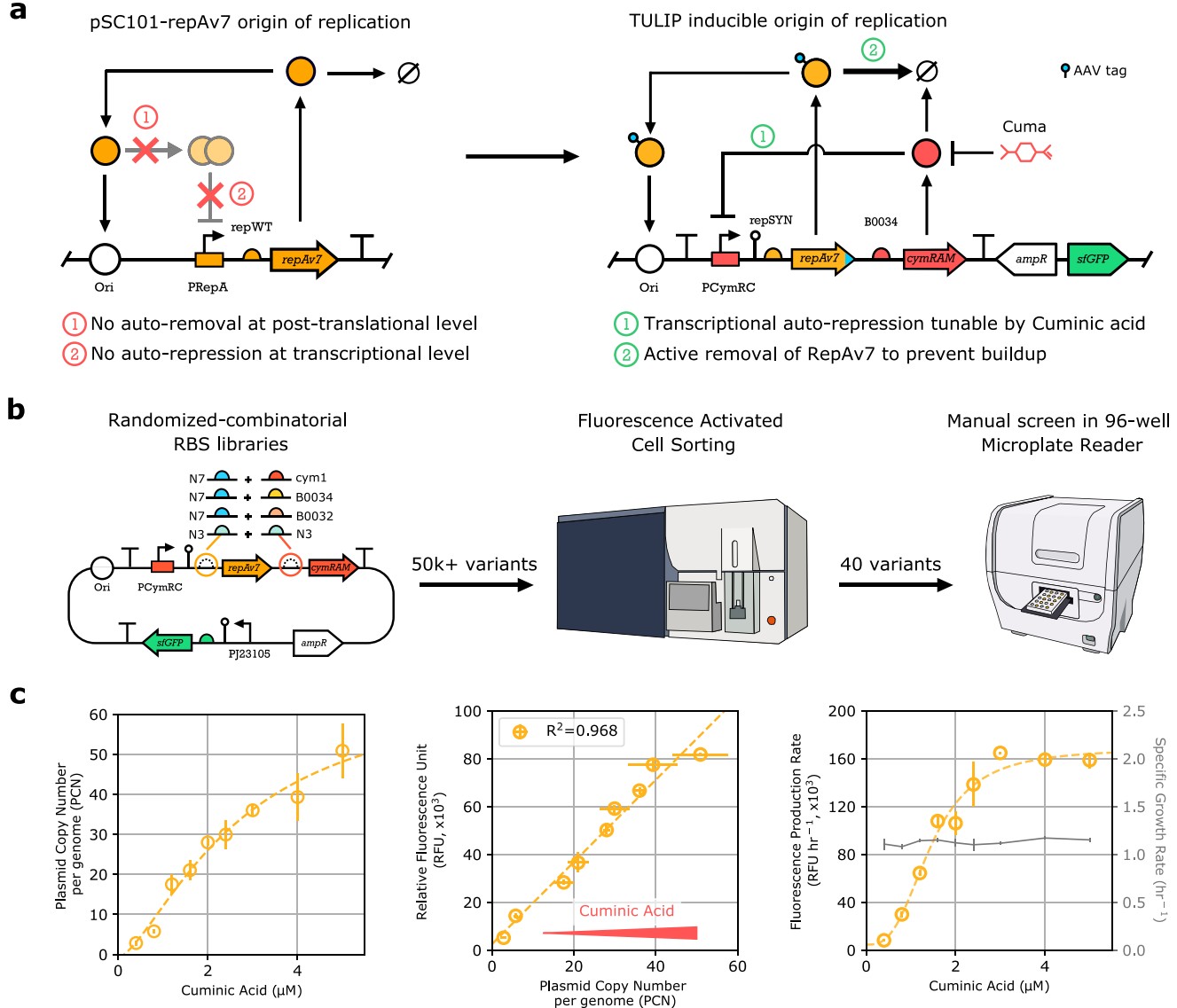

**Fig. 2 | Design, construction and screening of TULIP. a** Linear plasmid map of the mutant pSC101-repAv7 origin and of the final TULIP plasmid, illustrating the mechanism of PCN control. Upregulation of PCN is achieved via RepAv7, while negative feedback is realized via CymRAM and can be modulated via Cuminic acid. **b** Schematic visualization of the workflow for screening and selecting the final TULIP variant, including library design, fluorescent activated cell sorting, and manual screening with microplate reader. **c** TULIP was characterized in detail in NEBStable. PCN (measured via qPCR) and gene expression (measured via flow cytometry) show a strong linear relationship. Experiments in a microplate reader reveal that varying PCN via Cuminic acid has negligible impact on growth rate (gray). Cuminic acid induction levels are 0.4, 0.8, 1.2, 1.6, 2.0, 2.4, 3.0, 4.0, and 5.0 μM.

can control PCN relying on a two-plasmid architecture. To ensure self-contained inducible PCN control, we next integrate this control scheme on a single plasmid.

**Implementing inducible PCN control on a single plasmid**

Inspired by the WT pSC101 origin of replication (Fig. 1a), we designed a synthetic variant equipped with tunable auto-regulatory control (Fig. 2a), harboring all the required regulatory components. In particular, RepAv7 and the inducible CymRAM inhibitory TF are placed in a polycistronic configuration under the PCymRC promoter. This way, PCN is upregulated by the monomeric form of RepAv7, while the negative feedback through the dimerized form is effectively eliminated (due to the severely diminished dimerization capacity of RepAv7). Instead, downregulation of PCN is now realized via CymRAM. Furthermore, this effect can be modulated via Cuminic acid: upon its addition, the repression of RepAv7 by CymRAM is relieved, yielding greater expression of the former, eventually leading to increased

plasmid replication. The equilibrium is reached once the positive effect of RepAv7 is balanced by the negative effect of CymRAM on RepAv7 expression, determining the PCN of the architecture in Fig. 2a.

As the emergent PCN control depends on the delicate balance of the above two forces, we next optimized the translation rates of both RepAv7 and CymRAM. To this end, we constructed four plasmid libraries by introducing random mutations in the RBS of RepAv7 (N7 or N3) and of CymRAM (cym1, B0034, B0032 or N3), together with a constitutive sfGFP cassette to serve as a proxy for PCN quantification using fluorescence-based screening (Fig. 2b). The library assemblies were transformed into NEBStable cells and characterized in a Fluorescence Activated Cell Sorter (FACS). As the N7-B0034 library displayed the greatest diversity (Supplementary Fig. 2), it was selected for further processing. First, we performed four additional rounds of FACS screening to identify cells with low sfGFP expression in the absence of Cuminic acid (Supplementary Fig. 3), then 40 colonies were subsequently selected manually. Relying on microplate reader experiments

to characterize the growth rate and the responsiveness of these colonies to Cuminic acid (indicating inducible PCN behavior), ten variants were selected (Supplementary Fig. 3) for further characterization using a range of Cuminic acid concentrations.

The population distribution of fluorescence intensity displayed considerable variation for all selected colonies, especially at low Cuminic acid concentrations (Supplementary Fig. 4), suggesting potential excess RepAv7 accumulation. To increase the removal rate of RepAv7, we next characterized the effect of degradation tags fused to RepAv7 on the population distribution of fluorescence intensity. Two moderate-weak degradation tags were selected and tested using one of the colonies with a wide dynamic range: ASV and AAV[34]. The variants with these tags displayed comparable performance with significantly reduced population-level variance in the distribution of fluorescence intensity at low concentrations of Cuminic acid (Supplementary Fig. 5). For the final design of TULIP, the variant with the AAV tag was selected.

Following the optimization and screening of the single-plasmid design of TULIP, we characterized the effect of Cuminic acid induction on PCN, sfGFP expression, and growth rate (Fig. 2c). Relying on qPCR measurements, the PCN of TULIP can be modulated between 2.9 ($\pm$0.7) and 50.9 ($\pm$6.8) in NEBStable cells. Furthermore, sfGFP production correlates linearly with PCN echoing our observations for the fixed copy variants in Fig. 1b. Finally, peak growth rate showed no appreciable dependence on PCN induction, highlighting that while TULIP offers inducible PCN control, this does not come at the price of reduced cell fitness.

## TULIP is portable across a variety of *E. coli* strains

TULIP is designed and implemented in a self-contained fashion (Fig. 2a) to enable the plug-and-play expression and testing of gene circuits in a wide variety of *E. coli* strains. This way flexible PCN control is achieved without relying on a specific strain, or on additional DNA plasmids expressing a plasmid replication factor or other control protein[21–23,25]. To verify multi-strain portability of TULIP, we further characterized its behavior considering four commonly used *E. coli* strains (Fig. 3a) spanning several application domains including plasmid cloning (DH10B), protein expression (NEBExpress), metabolic engineering (BW25113), and microbiology model strains (MG1655), in addition to NEBStable (Fig. 2).

Following transformation, induction with Cuminic acid in all these strains confirmed via qPCR experiments that TULIP enables flexible PCN control independent of the host (Fig. 3b), and flow cytometry revealed that the previously observed approximately linear relationship between PCN and constitutive sfGFP expression is also retained (Fig. 3c). In addition, relying on microplate reader experiments, we concluded that induction of TULIP imposes negligible metabolic burden on the host as its growth rate remains practically unaffected (Fig. 3d). These results verify that TULIP can be successfully deployed for flexible PCN control across a wide range of *E. coli* strains (and also in different medias commonly used in experimental microbiology, such as M9-Glucose, M9-Glycerol, Lysogeny Broth, and Super Optimal Broth, as illustrated in Supplementary Fig. 7), though quantitative performance may display considerable strain-to-strain differences[35] that increase with phylogenetic distance (see Supplementary Section 1.3 for more details). For instance, TULIP shows a hypersensitive response to Cuminic acid in NEBExpress, a B-strain of *E. coli*, especially when compared to the progressive and regularly interspaced PCN control observed in the other K-12 strains. This may stem from the absence of several protein degradation machineries in NEBExpress (rendering B-strain *E. coli* the preferred host for protein expression), likely contributing to the excess accumulation of RepAv7. This in turn leads to elevated PCN at low Cuminic acid concentration and to the hypersensitive response to increasing levels of the inducer, though this issue can be easily mitigated by considering a finer concentration gradient of Cuminic acid (Supplementary Fig. 8).

## PCN can be dynamically regulated and reliably maintained using TULIP

The design and implementation of TULIP promises flexible and reliable PCN regulation over time by simply altering the Cuminic acid concentration of the growth media. To verify this, combining the Chi.Bio turbidostat platform[36] with flow cytometry analysis we next (i) demonstrated that PCN can be dynamically regulated; and (ii) characterized plasmid stability.

To demonstrate that multiple setpoint changes can be achieved with the same cell population, continuous culture experiments were carried out using the Chi.Bio turbidostat platform[36] to maintain constant cell density and to monitor sfGFP expression for over 50 generations. The fluorescence data in Fig. 4a reveal that PCN can be adjusted and maintained via Cuminic acid considering both low-to-high and high-to-low concentration transitions over a wide range, without noticeable effect on growth rate. Single-cell level analysis of samples taken prior to each transition further confirms population-level uniformity. Population-level data in Fig. 4a together with single-cell flow cytometry analysis in Supplementary Figs. 9a and 10a further reveal that PCN can be adjusted in approximately 3–4 h (corresponding to 3–5 generations) at all tested Cuminic acid induction levels.

Regarding plasmid stability, both population-level and single-cell flow cytometry data in Supplementary Fig. 9b, c confirm that TULIP can be reliably maintained for extended periods of time (over 50 generations) both at low and high PCN setpoints with negligible temporal variation. These results verify that over timescales typical in synthetic biology experiments, TULIP can be deployed without encountering challenges related to plasmid maintenance (i.e., complete loss from the population, or toxic build-up due to runaway replication) in the presence of antibiotic selection, despite the random nature of plasmid segregation during cell division. In the absence of antibiotic selection pressure, however, TULIP is gradually lost, a phenomenon that is particularly pronounced at low Cuminic acid concentration (Fig. 4b).

## TULIP reduces our reliance on cloning and transformation

By mitigating our reliance on time-consuming and error-prone cloning and transformation steps, TULIP accelerates the design and optimization of genetic modules. We illustrate this by considering two biochemical sensors responding to either 3OC6-HSL (AHL) or Vanillic acid (Van). In both of these modules (Fig. 5), a chemically inducible TF is constitutively expressed from the plasmid, activating transcription at its target promoter upon binding to its cognate ligand to express the reporter mScarlet-I (mScarI). As demonstrated in Fig. 5, varying Cuminic acid induction changes the PCN of TULIP harboring the sensor modules, thus we can implement a large variety of transfer curves without creating a combinatorial library of variants relying on cumbersome cloning and transformation tasks, alongside with time-consuming screening.

By opening up a third dimension to gene expression control, complementing transcriptional and translational regulation, TULIP can also be leveraged to characterize the impact of metabolic burden, e.g., when scarce resources need to be redirected from the expression of one gene upon induction of another[37–39]. The resulting coupling is captured via "isocost lines" previously uncovered when expressing two proteins from plasmids with fixed copy numbers[39]. While revealing the effect of PCN on these isocost lines originally required multiple cloning and transformation steps, with TULIP we only need a single one of each. For the single-layer sensor modules in Fig. 5 we characterized how production of the output (mScarI) affects the expression of the constitutively expressed reporter (sfGFP). In both cases, the production rate of the latter sharply decreases as activation of the former increases, recovering the isocost lines and their dependence on PCN: increasing Cuminic acid results in a shift upwards. Thus, modules harbored in TULIP not only recapitulate the expected input-output

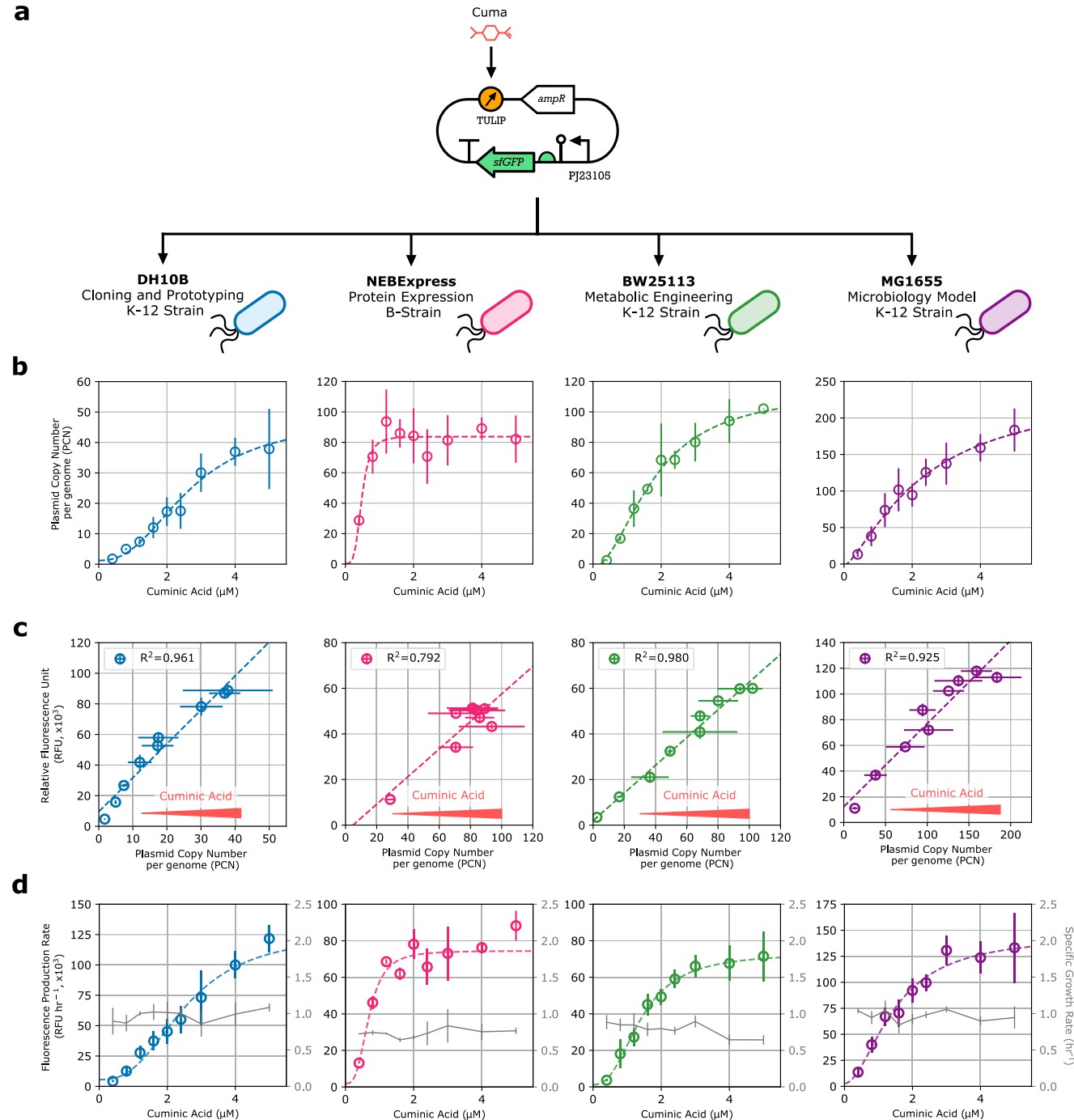

**Fig. 3 | TULIP enables flexible PCN control with multi-strain portability.**
Cuminic acid induction levels are 0.4, 0.8, 1.2, 1.6, 2.0, 2.4, 3.0, 4.0, and 5.0 μM.
**a** TULIP was transformed into a collection of commonly used *E. coli* strains, then induced with a range of Cuminic acid concentrations. **b** TULIP shows robust amplification of PCN across all strains tested (measured via qPCR). **c** PCN

(measured via qPCR) and gene expression (measured by flow cytometry) show strong linear correlation, as seen for the fPCN collection (Fig. 1) and for TULIP in NEBStable (Fig. 2). **d** Fluorescence production rate and growth rate (gray) were monitored in microplate reader experiments.

transfer curves, but also subtle features of resource competition. Despite the significant reduction in sfGFP expression, the PCN of both TULIP and plasmids with fixed copy numbers display similar and minor variation as a result of increased resource competition (Supplementary Fig. 13).

### TULIP promotes the re-use of modules in different contexts
In addition to accelerating the design and optimization of genetic modules, TULIP also facilitates their re-use in a context different from

where they were originally developed and characterized. Leveraging TULIP, here we reveal not only how PCN control facilitates design and optimization in a primary strain (DH10B), but also how a successful variant can be quickly re-used in a secondary strain (BW25113) equipped with a different genetic background.

We illustrate this by focusing on the sensor modules from Fig. 5. An implementation is considered successful if the ON state output exceeds a pre-defined threshold value (dashed gray lines in Fig. 6). To identify suitable candidates in the primary strain, one may synthesize a

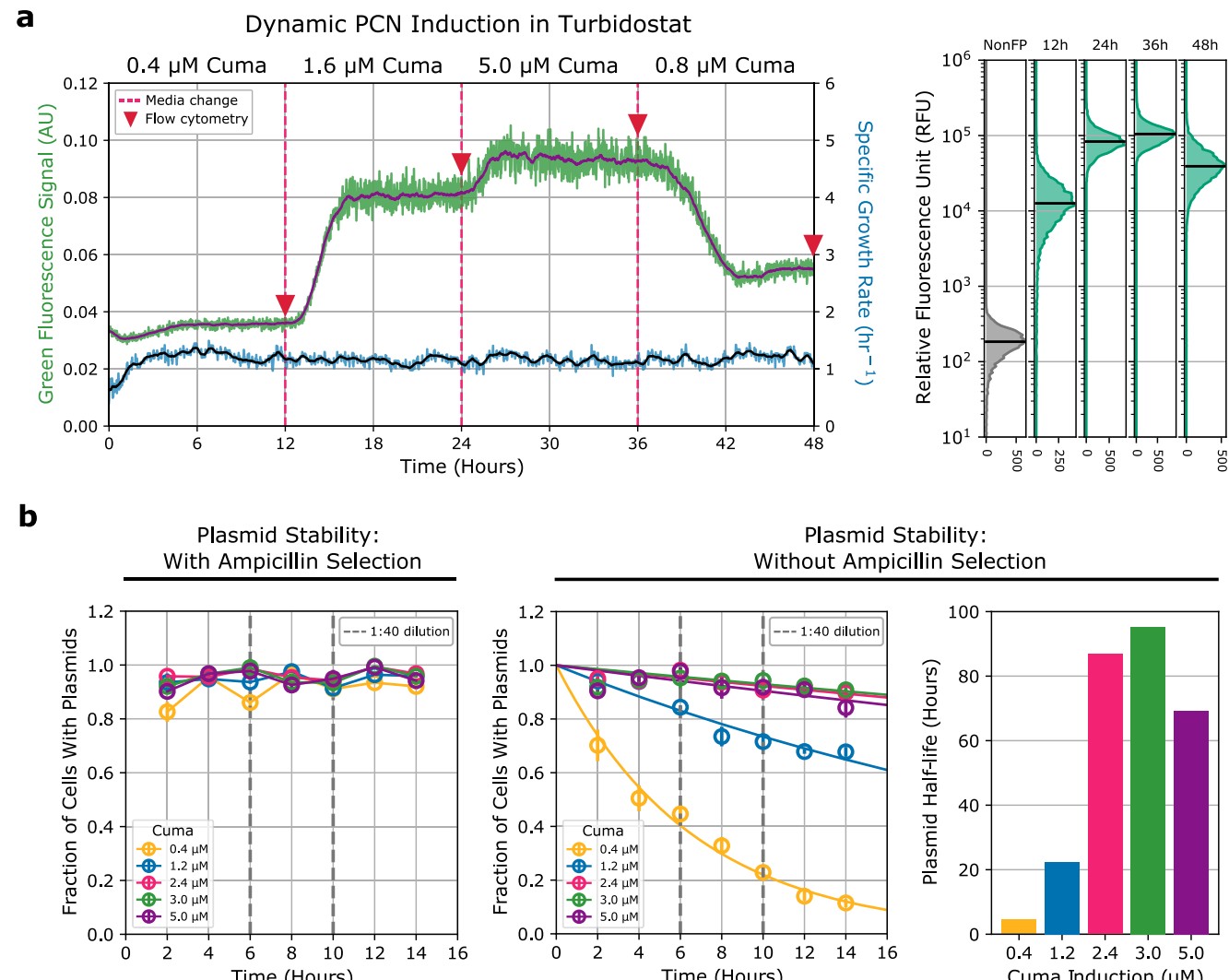

**Fig. 4 | TULIP enables dynamic and reliable PCN control.** All experiments were performed in DH10B, in M9-Glucose media. Non-fluorescent cells carrying no plasmid (NonFP) act as negative control. **a** TULIP was grown using the Chi.Bio turbidostat platform to maintain constant cell density. Fresh media contained varying levels of Cuminic acid to modulate PCN dynamically in the same cell population over time. Samples for flow cytometry analysis were taken prior to media changes every 12 h. Green and blue lines represent raw data recorded by the Chi.Bio user interface, purple and black line overlays represent 30-point moving average to highlight the key features of the trace data. **b** Plasmid stability with and without Ampicillin selection. Plasmid half-life was calculated assuming exponential decay. For more details, see Supplementary Figs. 10b and 12.

library containing a large number of candidate circuits (e.g., by varying the promoter and RBS regions), then clone each candidate into a collection of plasmids with fixed copy number, and screen this combinatorial library to find the right variant (with the correct promoters, RBSs, and fixed copy number). Instead, with TULIP each candidate needs to be cloned only once, and the PCN range can be swept efficiently by varying the Cuminic acid concentration.

This is demonstrated in Fig. 6a, revealing that when LuxR is expressed from a moderate-weak RBS (B0032) in DH10B, PCN needs to exceed approximately 28 to achieve our target ON state, whereas with a strong RBS (B0034) approximately 6 copies per cell is sufficient. Unfortunately, successful constructs in the primary strain (e.g., harboring the B0032 variant in a plasmid with PCN of 30, and the B0034 variant in a plasmid with PCN of 10) may not necessarily work in the secondary strain (Fig. 6a). Relying on plasmids with fixed copy number, this would require us to re-start the library preparation, cloning, transformation, and screening in the secondary strain. Instead, as our model-based analysis reveals that the sensor output monotonically increases with PCN (Supplementary Section 3.1), with TULIP it is sufficient to adjust Cuminic acid induction to ensure that PCN is

approximately doubled, without re-designing the sensor modules, confirmed in Fig. 6a for both variants. While there is no guarantee that changing solely the PCN is always sufficient, by drastically reducing cloning and transformation steps, TULIP accelerates the design and optimization of genetic modules and facilitates their re-use in different contexts.

Adjusting the PCN via TULIP is especially beneficial when the circuit may function properly only within a narrow range. According to our model-based analysis (Supplementary Section 3.1), this is precisely the case with the sensor module in Fig. 6b. In particular, we expect that while expressing VanRAM using a moderate-weak RBS (B0032) would result in a monotone Cuminic acid-mScarI relationship, increasing PCN via Cuminic acid can have a negative impact on mScarI when relying on a strong RBS (B0034). This is confirmed in Fig. 6b: although the former variant satisfies the sensitivity condition when the PCN exceeds 5 copies per cell, the latter produces satisfactory performance only within the narrow range of approximately 20–27 copies per cell. This highlights that finding the right PCN range could be of critical importance. While this task would require considerable cloning and transformation steps relying on plasmids with fixed copy number, with TULIP only a single

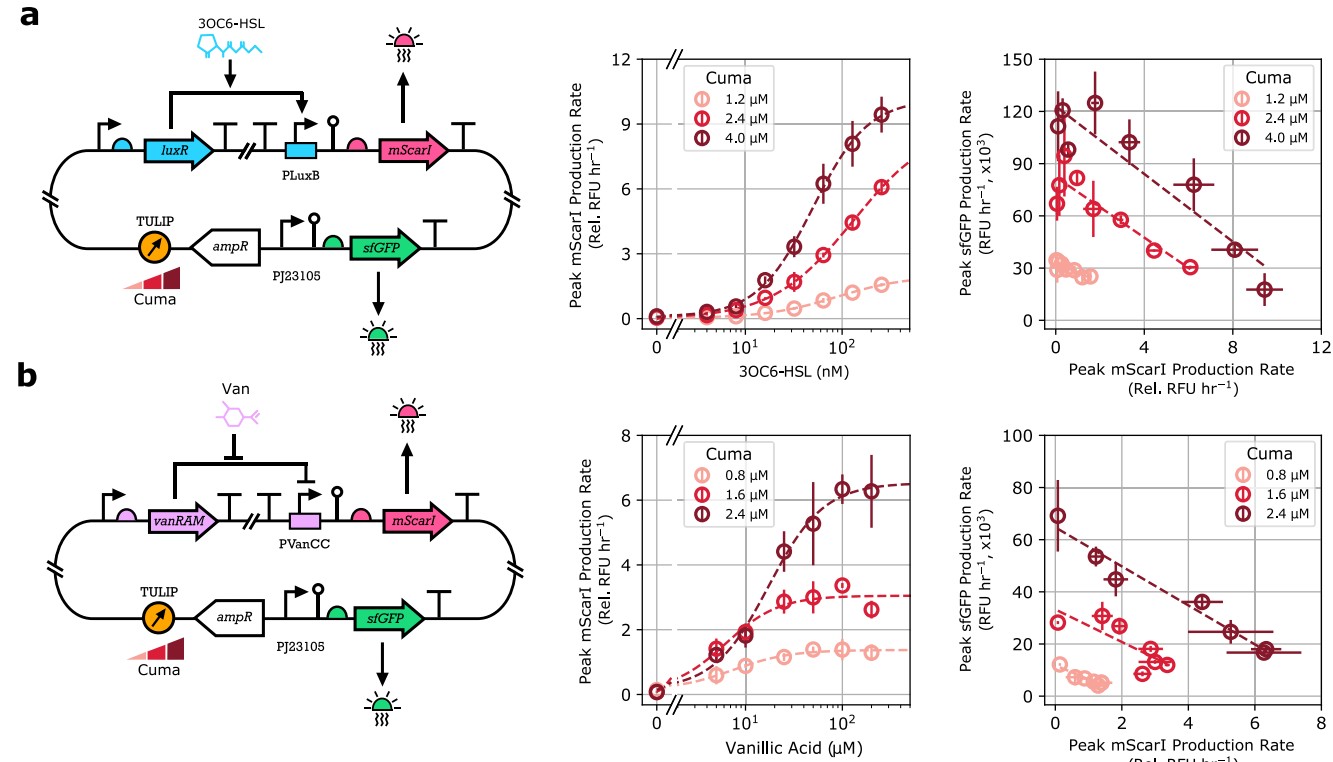

**Fig. 5 | TULIP accelerates the design and optimization of genetic modules via flexible PCN control.** All experiments were performed in DH10B. Light, medium, and dark red denote the varying Cuminic acid levels used to modulate PCN. Production of mScarI is normalized to DHScarI, a strain constitutively expressing mScarI, thus mScarI production is reported as a relative ("Rel.") value. **a** The AHL-LuxR transcription inducible circuit was cloned into TULIP. Expression of sfGFP and mScarI was quantified in a microplate reader for a range of AHL concentrations. **b** The Van-VanRAM transcription inducible circuit was cloned into TULIP. Expression of sfGFP and mScarI was quantified in a microplate reader for a range of Vanillic acid concentrations.

construct needs to be synthesized and tested at varying Cuminic acid induction levels. Furthermore, while the B0032 variant could be easily re-used in BW25113 by further addition of Cuminic acid to increase PCN approximately 5-fold, quickly sweeping through the possible PCN variants using TULIP reveals that the B0034 variant could likely not be re-used in this new host without modifying its parts.

Importantly, a similar approach can be taken with more complex circuits to optimize their strain-dependent performance and also to re-use them in different hosts, as we illustrate in Supplementary Fig. 14. However, while we expect TULIP to offer a valuable dimension to tune the behavior of genetic circuits irrespective of their size, the beneficial impact of flexible PCN control will likely have to be complemented with other regulatory and cloning strategies in case of increasing circuit complexity due to the quickly growing number of strain-dependent biophysical parameters.

Finally, the data in Fig. 6 also reveal key characteristics of the genetic parts themselves. To illustrate this, note that the output in the OFF state increases with PCN in Fig. 6a, but it remains practically unaffected in Fig. 6b. This suggests that the PLuxB promoter has considerable leakiness, whereas the PVanCC promoter is instead tightly regulated with negligible basal expression. Therefore, while in case of the sensor in Fig. 6b it is advisable to choose a variant with moderate-weak RBS (such as B0032) to ensure multi-strain portability, the leakiness of the PLuxB promoter suggests that for the sensor in Fig. 6a a stronger RBS (such as B0034) offers superior performance with increased dynamic range.

**TULIP can facilitate gene circuit prototyping**

Modules developed in TULIP can also be moved to plasmids with fixed copy number to be later utilized in downstream applications. We illustrate this by focusing on the toggle switch from ref. 40, underpinned by considerably richer dynamics than the sensor modules in Figs. 5 and 6, with the potential for bistability.

We consider the standard realization of the toggle switch[41] with LacI (co-expressed with mKate2) and TetR (co-expressed with eGFP) repressing each other (Fig. 7a). Therefore, addition of IPTG/aTc results in upregulation of eGFP/mKate2 and downregulation of mKate2/eGFP, respectively. To ensure that the steady state concentration of LacI/TetR match the input dynamic range of further downstream modules, the protein expression levels can be modified via the genetic parts themselves (e.g., by adjusting promoter and RBS strengths); however, this combinatorial library-based approach requires extensive cloning and transformation. Alternatively, using TULIP we may tune them by controlling the PCN via the addition of Cuminic acid without modifying the genetic layout, and once the correct PCN has been identified, clone the construct into the appropriate plasmid with matching fixed copy number.

To illustrate this, the genetic toggle switch in Fig. 7a was cloned into TULIP and into plasmids with fixed copy number spanning the same PCN range in DH10B (Fig. 7b). After overnight growth of 12 h, cells were induced with either IPTG or aTc, then the first round of samples were collected for flow cytometry analysis after 8 h (pre-wash, circles in Fig. 7c). As expected, cells induced with IPTG/aTc had elevated levels of eGFP/mKate2 expression, and this shift increased with PCN. To test stability of these induced steady states (and thus the bistability of the toggle switch at different PCN), following 4 more hours of growth cells were diluted and grown for 12 h in fresh media lacking both IPTG and aTc, then diluted again into fresh media before collecting the second round of samples for flow cytometry analysis after 8 h (post-wash, triangles in Fig. 7c).

From a qualitative perspective, the behavior of the toggle switch when harbored in plasmids with fixed copy number closely matches

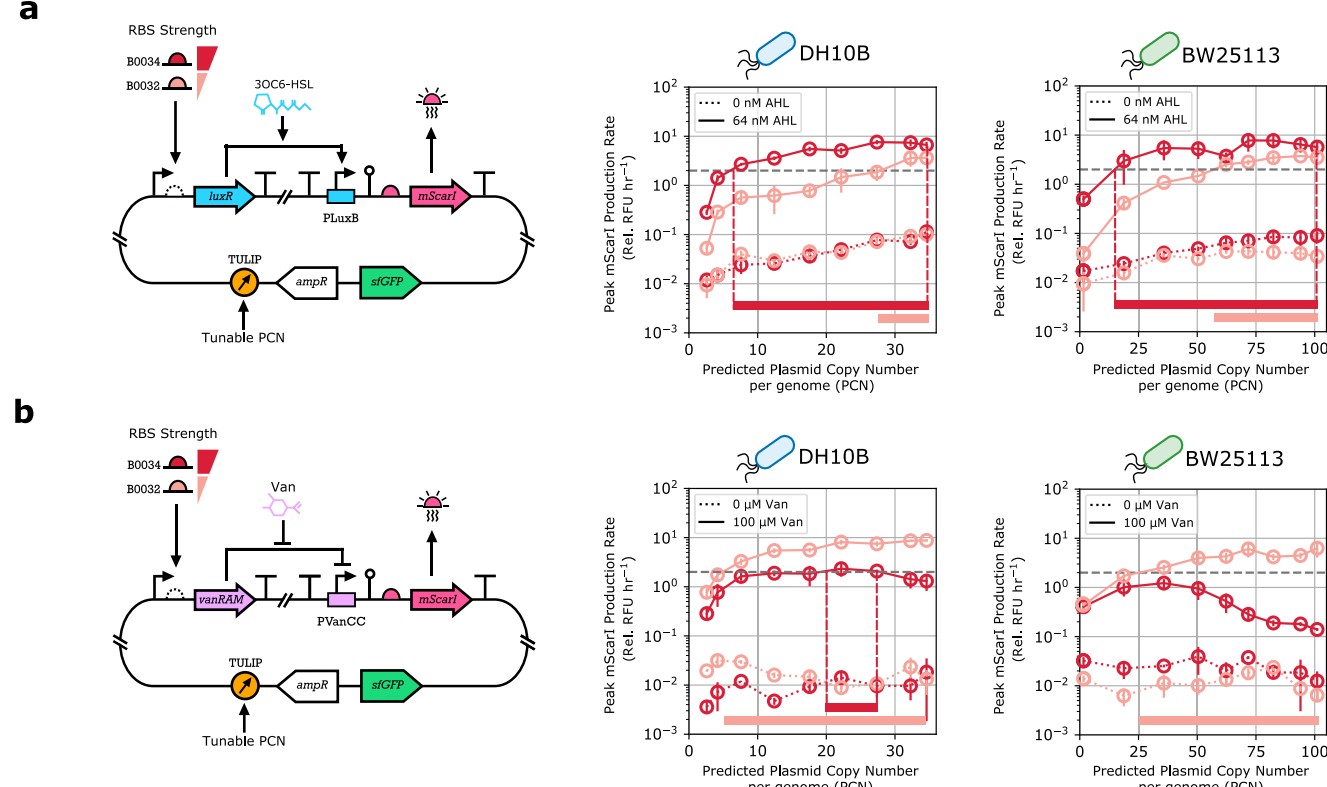

**Fig. 6 | TULIP enables the re-use of modules in different contexts via flexible PCN control.** Gene expression was quantified in a microplate reader. Induction via AHL (64 nM) or Van (100 μM) corresponds to the ON state, OFF state refers to the absence of these inducers. Cuminic acid induction levels are 0.4, 0.8, 1.2, 1.6, 2.0, 2.4, 3.0, 4.0, and 5.0 μM; the values for PCN are determined by the fits generated in Fig. 3b between Cuminic acid concentration and PCN quantified by qPCR. Production of mScarI is normalized to DHScarI and BWScarI, respectively, which are strains constitutively expressing mScarI, thus mScarI production is reported as a relative ("Rel.") value. **a** The AHL-LuxR sensor from Fig. 5a utilizing either B0032 (light red) or B0034 (dark red) was cloned into TULIP, then transformed into DH10B and BW25113 strains. Horizontal bars indicate the PCN range where the output in the ON state exceeds the critical threshold denoted by the dashed gray line. **b** The Van-VanRAM sensor from Fig. 5b utilizing either B0032 (light red) or B0034 (dark red) was cloned into TULIP, then transformed into DH10B and BW25113 strains. Horizontal bars indicate the PCN range where the output in the ON state exceeds the critical threshold denoted by the dashed gray line.

that observed when using TULIP, as eGFP and mKate2 expression (both pre-wash and post-wash) follow the same trend independent of the backbone (Fig. 7c). Furthermore, while pre-wash and post-wash samples are nearly identical at medium and high PCN levels, there is significant decrease of eGFP when PCN becomes sufficiently low, yielding almost identical post-wash concentrations subsequent to IPTG and aTc induction in case of both types of backbones. In addition to these qualitative similarities, data obtained using TULIP accurately predict the behavior observed when relying on plasmids with matching fixed copy number (Fig. 7d) from a quantitative perspective as well (using sfGFP expression from the empty backbone as a proxy for PCN). Finally, as the eGFP decrease in post-wash samples at low PCN values subsequent to IPTG induction is likely a result of low basal expression of LacI from the chromosome in the original DH10B strain, we repeated the experiments with this gene knocked out. As expected, this modification had identical impact at low PCN levels (reduced drop in post-wash eGFP expression subsequent to IPTG induction), and more importantly, preserved the close quantitative alignment between the behavior observed when using the two types of backbones (Supplementary Fig. 20).

These findings demonstrate not only that PCN control offers a simple and convenient way to tune circuit dynamics (e.g., observed steady states of the toggle switch span almost a 20-fold range), but also that the type of the backbone (TULIP or plasmids with fixed copy number) has minimal impact on the observed behavior. Thus, TULIP can facilitate gene circuit prototyping via flexible and dynamic adjustment of PCN without relying on time-consuming and error-prone cloning and

transformation steps to guide the selection of parts and conditions (e.g., inducer concentration, PCN range) to ensure correct functioning (e.g., desired stability profile, dynamic range) even when modules are eventually deployed in plasmids with matching fixed copy number. Importantly, however, while TULIP provides us with a powerful dimension for tuning circuit behavior, it complements rather than replaces tuning via combinatorial libraries[42]. For instance, a toggle switch relying either on monomeric repressors or on unbalanced production rate constants may not be rendered bistable by only modifying the PCN of the plasmid harboring it (Supplementary Fig. 106).

## TULIP enables versatile PCN control using a variety of input stimuli

While TULIP is originally developed to implement Cuma-responsive PCN control relying on a single plasmid, in Fig. 8a we illustrate that it can be easily interfaced with an additional layer of control to further increase its versatility. Within this scheme, PCN is modulated via CRISPR interference (CRISPRi): a small guide RNA (gRNA) targeting the CymRAM coding sequence recruits dCas9 to implement sequence-specific repression of CymRAM expression. Increased gRNA concentration hence plays a role similar to induction via Cuminic acid as they both relieve the repression of the PCymRC promoter: the former via inhibiting the expression of CymRAM, the latter via preventing it from binding to the promoter. Thus, by coupling gRNA expression to the signal of interest (e.g., chemical inducer, protein concentration, environmental factors), a wide variety of input stimuli can regulate PCN by interfacing this additional layer of control with TULIP. This way,

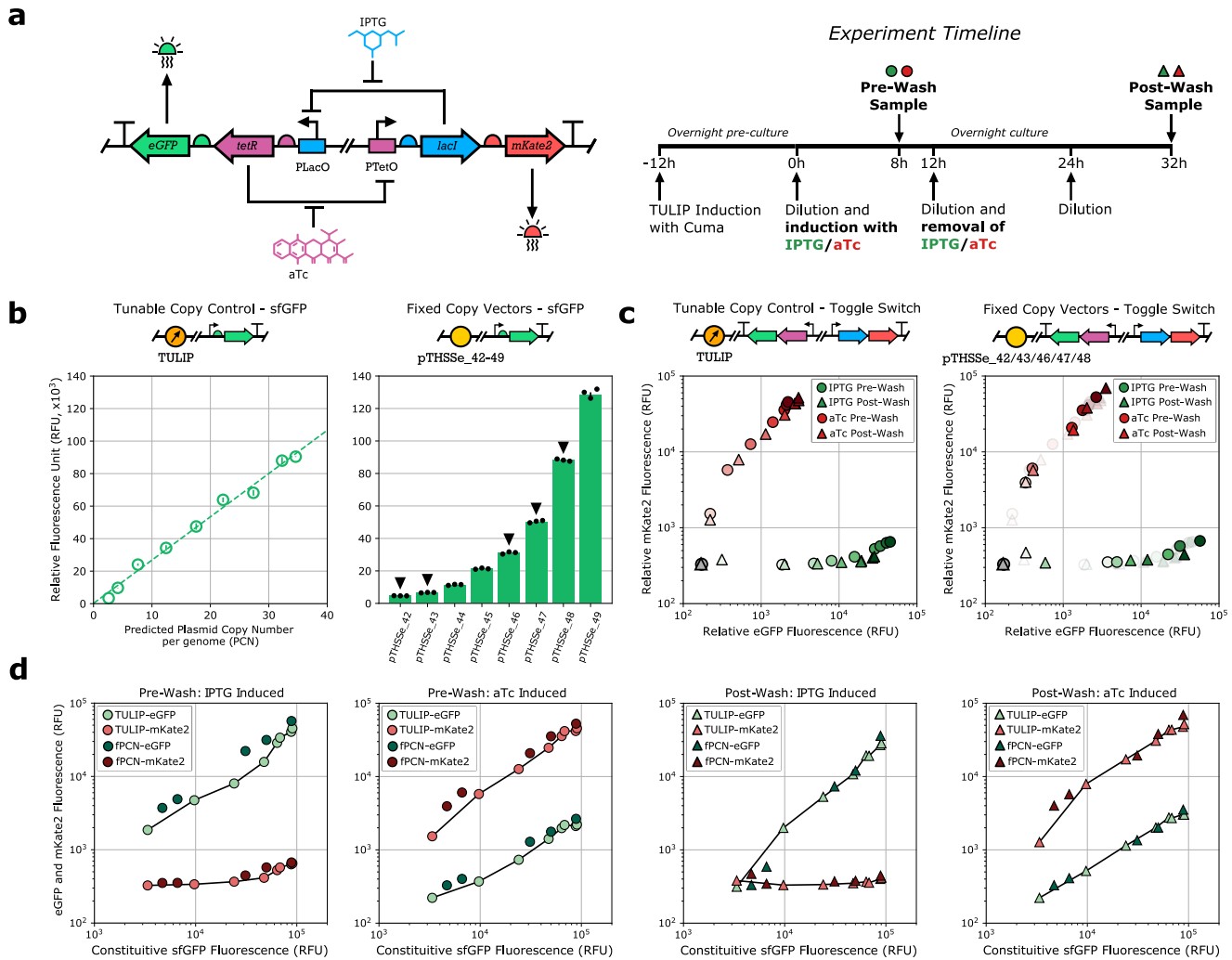

**Fig. 7 | TULIP can facilitate gene circuit prototyping.** LacI is co-expressed with mKate2, TetR is co-expressed with eGFP. Additional flow cytometry data and analysis are provided in Supplementary Figs. 14–19. Circles and triangles denote pre-wash and post-wash samples, respectively. **a** Genetic layout of the toggle switch from ref. 40 alongside with the experimental timeline for collecting samples subsequent to induction with IPTG/aTc (pre-wash, circles) and following their removal from the media (post-wash, triangles). **b** Expression of sfGFP from the empty backbone (proxy for PCN). PCN estimates for TULIP are obtained via qPCR experiments. Among plasmids in the fPCN collection, those indicated with black arrowheads were selected to harbor the toggle switch. **c** Behavior of the toggle

switch when harbored in TULIP (PCN is modulated via the addition of Cuminic acid at the following concentrations: 0.4, 0.8, 1.2, 2.0, 2.4, 3.0, 4.0, and 5.0 μM) and when harbored in plasmids with fixed copy number (pTHSSe_42, pTHSSe_43, pTHSSe_46, pTHSSe_47, pTHSSe_48). Green/red indicates induction with IPTG/aTc. Darker shades indicate higher PCN, gray corresponds to empty cells with no fluorescent constructs, symbols in the background denote data obtained with TULIP. **d** Pre-wash and post-wash eGFP (green) and mKate2 (red) expression levels as a function of sfGFP expression from the empty backbone (proxy for PCN), both in case of TULIP (light symbols) and plasmids with fixed copy number (dark symbols).

the application-specific module connecting the input of interest to gRNA expression can be easily swapped without further screening or tuning, instead, we leverage the carefully balanced feedback loops that TULIP is already equipped with.

To illustrate this approach, we designed eight gRNAs targeting the CymRAM coding sequence (gRNAs #1–#8) alongside with a negative control (gRNA #9), each expressed from either the Van-inducible PVanCC promoter or the AHL-inducible PLuxB promoter. In media supplemented with 0.4 μM Cuminic acid, while gRNA #9 expression had no impact on PCN, upregulation of all other gRNAs resulted in increased PCN, verifying the correct functioning of the proposed control scheme. Selecting the gRNA design yielding the most prominent PCN response (gRNA #5), data in Fig. 8b, c verify that PCN monotonically increases with the inducer concentration in both realizations, in spite of the low Cuminic acid concentration (using gRNA #6 yields similar performance, see Supplementary Fig. 26). Furthermore, the PCN range spanned in this extended design in response to AHL/

Van closely matches the one obtained as a result of Cuminic acid induction (gray shaded regions in Fig. 8b, c) without any adjustments in either TULIP or the CRISPRi-based control modules. Both implementations in Fig. 8b, c rely on the same gRNA and required no application-specific tuning of either the gRNA or the module regulating its expression. This highlights that the increased versatility of the regulatory scheme in Fig. 8 does not come at a price of decreased performance, further illustrating that TULIP can be deployed and utilized in a diverse array of synthetic biology applications. While in this proof-of-concept implementation we rely on the FR-E01 strain for dCas9 expression[43], this gene can also be integrated into the plasmid harboring the additional layer of control to ensure multi-strain portability even of this expanded control scheme.

## Discussion
Gene expression control plays an essential role in the advancement of biological research, especially in the field of synthetic biology.

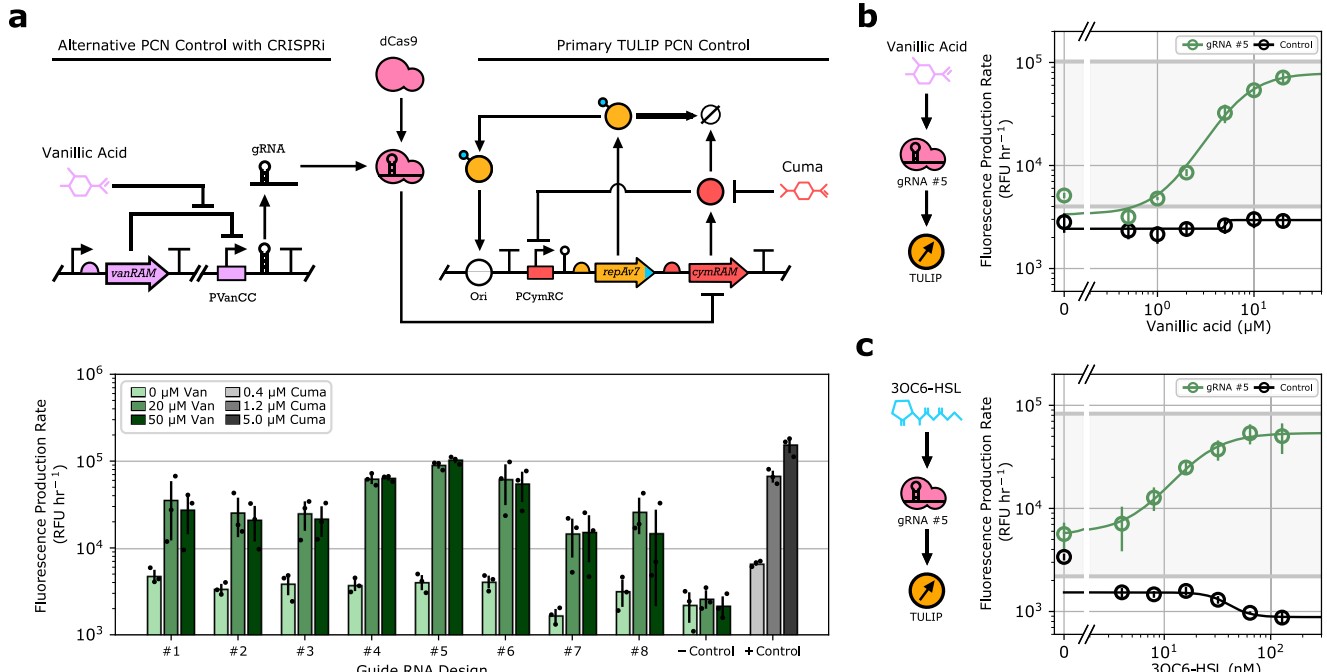

**Fig. 8 | TULIP enables versatile PCN control using a variety of input stimuli.** The empty TULIP backbone is co-transformed with the gRNA expression cassette harbored on a p15A-Kan plasmid into the FR-E01 strain expressing dCas9 upon aTc induction[43]. Expression of sfGFP is measured in microplate reader experiments. All conditions include 0.4 μM Cuminic acid for minimum, stable PCN maintenance. Each experiment was conducted in biological triplicates (*n* = 3), data points and error bars represent the mean and standard deviation across replicates. **a** Genetic layout of the gRNA-based module providing an alternative mode of PCN control.

Negative and positive control refer to a non-targeting gRNA #9, and to TULIP induced with Cuminic acid (gray bars), respectively. **b** Expression of the gRNA is regulated via induction of the PVanCC promoter using Vanillic acid. The gray shaded region indicates TULIP's functional range corresponding to 0.4–5.0 μM Cuminic acid induction. **c** Expression of the gRNA is regulated via induction of the PLuxB promoter using AHL. The gray shaded region indicates TULIP's functional range corresponding to 0.4–5.0 μM Cuminic acid induction.

Allowing easy and efficient exploration of the gene expression design space, a plethora of tools enable flexible and dynamic control at the transcriptional and translational levels. The versatility of these techniques highlights a critical gap in our toolkit as control at the DNA plasmid copy number level is severely limited: current approaches either rely on plasmids with fixed copy numbers[44], or require multiplasmid constructs[24–26], or are restricted to select strains[21–23]. To address this challenge, with TULIP we extended flexible gene expression control into the third dimension via PCN regulation, complementing transcriptional and translational tools.

TULIP is a self-contained DNA plasmid that offers inducible PCN control spanning a wide dynamic range and performs robustly across multiple commonly used *E. coli* strains and growth media. As we demonstrated through multiple illustrative examples, by mitigating our reliance on time-consuming and error-prone cloning and transformation steps underpinning current approaches, TULIP accelerates performance optimization of gene circuits, the re-use of modules in different contexts, and facilitates gene circuit prototyping. Furthermore, it can be leveraged to probe how competition for shared cellular resources affects the behavior of seemingly unconnected circuit components, and to reveal crucial properties of genetic parts that would be difficult to characterize using plasmids with fixed copy number. As the expression of genes harbored in TULIP can be dynamically adjusted not only via Cuminic acid but also via a variety of input stimuli through the additional layer of CRISPRi-based control that we have developed, our platform offers a convenient way to modulate circuit activity and to ensure robust performance despite dramatic shifts in the cellular context[45].

Recent advances highlight that dynamic pathway regulation via PCN control can improve the performance of microbial cell factories[26], and can also be leveraged to characterize the effects of PCN on cellular growth rates, gene expression, biosynthesis, and genetic circuit performance[23]. Complementing these approaches that significantly expand our synthetic biology toolkit, TULIP was designed with modularity and orthogonality in mind to enable its adoption with ease and minimal disruption to already existing systems. By using Cuminic acid as the primary mode of PCN control and the pSC101 origin from incompatibility group C as our scaffold (thus avoiding staple chemical inducers such as IPTG and aTc, and plasmids in incompatibility group A and B such as pColE1, pBR322, pUC19, and p15A), TULIP is cross-compatible with a wide array of commonly used vectors and genetic components. In addition, with its demonstrated robust functionality considering a range of *E. coli* hosts, media compositions, and synthetic gene circuits, TULIP offers a flexible platform that can be deployed effortlessly in vastly different contexts to mitigate known issues of synthetic gene circuits, such as narrow dynamic range in activation and repression control[46,47], leaky expression[48,49], and metabolic burden[23,50]. As we illustrate in Fig. 6, PCN control can be leveraged to increase the dynamic range of synthetic gene circuits[16,51,52], to reduce basal expression from leaky promoters, and to ensure that expression levels can be increased without suffering the adverse effects of the often unavoidable increased leakiness. In addition, as maintenance and replication of plasmids and the overexpression of recombinant proteins represent considerable metabolic burden[53,54], dynamic pathway regulation via PCN control can improve the performance of microbial cell factories[26], especially since low-copy plasmids can often outperform their high-copy counterparts in metabolic engineering[55].

In addition to the application examples we considered in this paper, we envision dynamic PCN control to be deployed in a wide range of settings. Potential contexts could include dynamic spatial organization of metabolic pathways with the plasmid acting as a scaffold to localize fusion proteins consisting of DNA binding domains

and metabolic enzymes[56,57], and alternative mechanisms for gene expression regulation, for instance, using plasmids to carry decoy operator sequences as a method for altering gene circuit dynamics and mitigating toxicity[58,59], or interacting with the host's native TFs for "soft" regulation to alter metabolism[60]. In addition, dynamic PCN control not only provides us with a straightforward way to simultaneously modulate the expression of multiple gene targets, it can also offer an economical alternative complementing transcriptional, translational, and post-translational control[37]. Finally, as Cuminic acid must be present for stable maintenance of TULIP (Supplementary Fig. 6), our synthetic DNA plasmid provides a promising avenue for biocontainment by mitigating risks of synthetic plasmid propagation and contamination in waste streams[61], and provides a simple mechanism to cure cells of plasmids with transient functions (such as in lambda-RED genomic integration[62] or in case of mutagenesis-inducing plasmids for in vivo directed evolution[63]).

Biology has evolved powerful and creative solutions to optimize gene expression control by harnessing regulatory mechanisms across multiple dimensions. With TULIP we rendered PCN control as accessible as regulation at the transcriptional and translational level to enable and encourage the flexible and efficient exploration of the design space in biological engineering from alternative perspectives.

# Methods

## Strains
A comprehensive list of all *E. coli* strains used in our work, along with their genotypes and source, are listed in Supplementary Table 1. NEBStable and NEB10Beta (referred to as DH10B) were used for all plasmid cloning purposes. NEBStable was used for the initial characterization of the fixed plasmid copy number variants (Supplementary Table 2) and for the two-plasmid inducible copy number system (Fig. 1). In addition, NEBStable was used for preparing the RBS library for screening TULIP, and as the host for subsequent characterization leading to the final design for TULIP (Supplementary Fig. 2). DH10B, NEBExpress, BW25113, and MG1655 were used for verifying multi-strain portability of TULIP (Fig. 3). DH10B was used extensively to characterize the stability of TULIP (Fig. 4 and Supplementary Figs. 7–13), and was used alongside BW25113 for characterizing the single-layer sensor modules (Figs. 5 and 6) and the genetic NOT-gate (Supplementary Fig. 14).

For the genetic toggle switch experiments (Fig. 7), DH10B was used as the host strain. In addition, we created a lacI deficient DH10B strain which was used to further validate the behavior of the genetic toggle switch. We generated a ΔlacI::kanR mutation on the genome using a modified protocol of the lambda-RED recombination system using the pREDCas9 construct[62,64]; pREDCas9 was a gift from Tao Chen (Addgene plasmid #71541)[65]. The primers used to generate the KanR genome integration cassette can be found in Supplementary Table 9.

For the CRISPRi experiments in Fig. 8, strain FR-E01 (K-12 MG1655 strain with a genomically integrated aTc-inducible dCas9 cassette) was used[43]; FR-E01 was a gift from David Bikard (Addgene plasmid #118727). Red fluorescent protein control strains were constructed by integrating a constitutively expressed mScarI cassette (J100mScarI: PJ23100_RiboJ_B0034_mScarlet-I_ECK120029600) into the genome using the pOSIP-CH system[66]. This mScarI cassette was integrated into the HK022 attB site in the DH10B and BW25113 strains resulting in DHScarI and BWScarI (Supplementary Table 1).

## Transformation protocols
For heat-shock transformation, the standard calcium chloride competent cell preparation protocol was used[67]. Electroporation was carried out using the protocol outlined in ref. 68, using a Bio-Rad MicroPulser Electroporator and 0.1 cm gap cuvettes (brown caps). Post transformation, NEBStable and DH10B cells were recovered in 1 mL NEBStable or NEB DH10Beta Out Growth media, respectively, all other

transformations were recovered in 1 mL SOB (Super Optimal Broth); transformations were generally incubated for 1 h at 37 °C, followed by plating on LB agar plates with appropriate antibiotic selection. For transformations with TULIP plasmids, the recovery media was supplemented with 1 μM Cuminic acid. Certain plasmids had temperature sensitive components, i.e., pOSIP-CH or pREDCas9, where appropriate these were always cultured at 30 °C.

## Media and growth conditions
Super Optimal Broth (SOB; yeast extract 5 mg/mL, L-tryptone 20 mg/mL, NaCl 10 mM, KCl 2.5 mM, MgCl$_2$ 10 mM, and MgSO$_4$ 10 mM) was used for all cloning and competent cell preparation purposes. Miller's Luria Broth (LB; also known as Lysogeny Broth) with agar (Sigma) was used for all solid medium plate cultures. Supplemented M9-Glucose media (referred to as M9-Gluc media: M9 salts 1X, thiamine hydrochloride 0.25 mg/mL, casamino acids 2 mg/mL, MgSO$_4$ 2 mM, CaCl$_2$ 100 μM, and glucose 4 mg/mL) was used for all experiments including flow cytometry, qPCR, microplate reader, and Chi.Bio turbidostat experiments. For the experiments presented in Supplementary Fig. 7, LB broth media was prepared using Miller's LB Broth (Sigma), and Supplemented M9-Glycerol media (M9-Gly) was prepared exactly the same as M9-Gluc, except a final concentration of 4 mg/mL glycerol was supplemented in place of glucose.

Unless otherwise indicated, all media and plates were supplemented with appropriate antibiotics to the following final concentrations: ampicillin at 100 μg/mL, kanamycin at 50 μg/mL, and chloramphenicol at 25 μg/mL. In experiments where double antibiotic selection was used, the final concentration of each antibiotic was halved in both plates and liquid cultures. For culturing cells carrying TULIP constructs, all cloning plates and liquid cultures were supplemented with 1 μM Cuminic acid for stable maintenance of TULIP, in addition to the appropriate antibiotic concentration. Chemical inducers Cuminic acid (4-Isopropylbenzoic acid), 3OC6-HSL (N-(3-Oxo-hexanoyl)-L-homoserine lactone; also referred to as AHL), Vanillic acid (4-Hydroxy-3-methoxybenzoic acid; also referred to as Van), IPTG (Isopropyl β-D-1-thiogalactopyranoside), and aTc (Anhydrotetracycline hydrochloride) were purchased from Sigma. All bacterial incubation was carried out at 37 °C. For liquid cultures, a shaking speed of 300 rpm was used for shaking incubators (New Brunswick Innova 42R, orbital diameter of 19 mm) and 800 rpm for plate shakers (Ohaus Incubating Microplate Shaker, orbital diameter of 3 mm).

## Cloning and plasmid assembly strategies
Cloning related enzymes and their respective reagents were sourced from New England Biolabs (NEB). Primers were ordered through Integrated DNA Technologies (IDT); Q5 Hot Start 2X Master Mix was used for all polymerase chain reactions (PCR; Bio-Rad C1000 thermal cycler), following the recommended protocol by NEB for a reaction volume of 25 μL. PCR reactions were verified by gel electrophoresis of 5 μL of the reaction volume on a 1% agarose gel (Sigma). The rest of the 20 μL PCR reaction volume was mixed with 1 μL of DpnI and digested for 1 h at 37 °C, followed by heat inactivation for 20 min at 80 °C. Post DpnI digest, PCR products were purified via the QIAquick PCR Purification Kit (Qiagen).

Plasmid construction was carried out using Golden Gate cloning[69]. Specifically, the CIDAR MoClo "sticky end" standard was used for all assemblies[70]; assembly "parts" were either used directly from MoClo plasmids, or were PCR amplified with BsaI cut sites and appropriate sticky ends designed into the primer overhangs. The CIDAR MoClo Parts Kit (Addgene kit #1000000059) was a gift from Douglas Densmore, and the CIDAR MoClo Extension Volume I Kit (Addgene kit #1000000161) was a gift from Richard Murray. For Golden Gate assembly reactions, 40 fM of backbone DNA and 80 fM of each insert DNA were mixed together, followed by 0.5 μL BsaI-HFv2 (20 units/μL),

0.5 µL T4 DNA Ligase (400 units/µL), 2 µL T4 DNA Ligase 10X buffer, and sterile MilliQ water to a final volume of 20 µL. The assembly mix would then be incubated in a thermal cycler with the following protocol: [5 min at 37 °C, 5 min at 16 °C] × 30 cycles, 30 min at 16 °C, 20 min at 80 °C, infinity at 10 °C. An assembly mix of 5 µL was then transformed to chemically competent NEBStable or DH10B cells via heat-shock transformation, followed by recovery in appropriate outgrowth media for 1 h, and then plating on two appropriate antibiotic plates for each reaction split in a 1:9 ratio, then incubated overnight at 37 °C. For minipreps, single colonies were inoculated into 5 mL of SOB media supplemented with appropriate antibiotics and cultured overnight, then miniprepped using the QIAprep Spin Miniprep Kit (Qiagen). A list of all plasmids and DNA parts used can be found in Supplementary Tables 2–8, and plasmid maps can be found in Supplementary Figs. 102 and 103.

### RBS library preparation, FACS and manual screening

The RBS libraries were assembled by relying on 2-part Golden Gate assembly using PCR fragments, as illustrated in Supplementary Fig. 2. The primers used to construct the library can be found in Supplementary Table 9. The assembly mix was transformed into chemical competent NEBStable cells and plated 100% on LB agar plates supplemented with ampicillin, and incubated overnight at 37 °C. The following morning cells were scraped off the plate and resuspended in approximately 2 mL filter-sterilized PBS. The cells were thoroughly mixed, then diluted 1:20 into a volume of 5 mL PBS. The samples were then processed by a BD FACSAria III Sorter, running on the BD FACSDiva v7.0 software. In the first screen approximately 100,000 cells were selected at the low fluorescence range from the library of choice, these were then plated on two LB agar plates and incubated overnight at 37 °C. The following morning cells were scraped and resuspended again in 2 mL of PBS, then diluted 1:20 into 5 mL of PBS and processed by the BD FACSAria III Sorter; this screen was repeated for a total of four rounds, and in screens #2–#4 a total of approximately 50,000 cells were gated per round. From the final plate after screen #4, 40 colonies with low fluorescence intensity were manually selected with the plate placed over a transilluminator (Biosystems iBright CL1500 imaging system). These colonies were subsequently cultured overnight in M9-Gluc media with either 0 or 1 µM Cuminic acid, then diluted 1:200 into fresh M9-Gluc media with the same inducer conditions and loaded into the microplate reader. Further details for the screening process can be found in Supplementary Section 1.2.

### General protocol for all experiments

Unless otherwise indicated, M9-Gluc media was used for all experimental overnight pre-cultures and for the experimental cultures themselves. For all experiments involving cells carrying TULIP, overnight pre-cultures were also induced with the appropriate concentration of Cuminic acid. Unless otherwise indicated, experimental cultures were carried out in a volume of 200 µL of M9-Gluc. "NonFP" control cells without any plasmid or fluorescence were used as negative control for fluorescence, and as positive control for growth. Cell strains with genomically integrated mScarI expression (DHScarI and BWScarI, Supplementary Fig. 103 and Supplementary Table 1) were used as positive control for red fluorescence in all experiments measuring mScarI expression, to which all calculated mScarI production rates were normalized. All experimental conditions (including controls) were done in biological triplicates, defined as 3 colonies from each strain to be tested. For all experiments in 96-well plates (deep-well and microplates) a breathable film (Sigma Breathe-Easy sealing membrane) was applied firmly with an ink roller.

### General quantitative PCR experiment protocol

Unless otherwise indicated, overnight pre-cultures were diluted 1:200 into fresh M9-Gluc media, with their respective inducer

concentrations, and cultured for 5 h at 37 °C, shaking at 800 rpm in 2 mL 96-well deep-well plates. At the time of sampling, 10 µL of cell culture was transferred to 90 µL of sterile MilliQ water in PCR tube strips, and the samples were boiled for 20 min at 95 °C in a thermal cycler. Samples were then stored at −20 °C until needed for quantifying PCN via quantitative PCR (qPCR).

For the qPCR experiments, Fast SYBR Green 2X mastermix (Fisher) was used and the reactions were run in a Biosystems QuantStudio 7 Pro with a 384-well module. Each biological sample was repeated in technical duplicates and each reaction had a final volume of 10 µL, composed of 5 µL enzyme mastermix, 0.25 µL of each forward and reverse primer (20 µM), 2.5 µL water, and 2 µL of the cell lysates. Primers to target the DXS gene[24] (one copy per genome) and the ampicillin resistance gene on the plasmid were created using the Primer3Plus design tool[71]. All designs are available in Supplementary Table 9. Primer efficiencies were calculated using a 5-fold dilution series gradient of either MG1655 genomic DNA (for the DXS primers), or an ampicillin resistant plasmid (for the ampicillin primers). Primer efficiencies of 101.4% (amplification factor 2.01) and 105.6% (amplification factor 2.06) were determined for the DXS and ampicillin primers, respectively. The Ct (threshold cycle) values were extracted from the raw fluorescence data for the DXS and ampicillin amplicons using the QuantStudio software (Design and Analysis v2.5), from which PCN was quantified[72] as

$$PCN = \frac{(\text{Amplification Factor DXS})^{CtDXS}}{(\text{Amplification Factor Amp})^{CtAmp}}.$$

All qPCR experiments also had triplicates of just mastermix, water, and each primer pair as negative controls.

### General flow cytometry protocol

Unless otherwise indicated, overnight cultures were diluted 1:200 into fresh M9-Gluc media, with their respective inducer concentrations, and cultured for 5 h at 37 °C, shaking at 800 rpm in 2 mL 96-well deep-well plates. At the time of sampling, 20–50 µL of cells were transferred to filter-sterilized PBS to a final volume of 200 µL in a 96-well microplate. The samples were then analyzed by a Biosystems Attune NxT with an autosampler attachment, running on the Attune™ NxT Software; at least 10,000 gated cell events were recorded per sample, and relative green fluorescence units (i.e., for sfGFP and eGFP) were quantified at Ex: 488 nm and Em: 530/30 nm (BL1), and relative red fluorescence (i.e., for mScarI and mKate2) were quantified at Ex: 561 nm and Em: 620/15 nm (YL2). All flow cytometry experiments had triplicates of NonFP cells as negative controls. An example of gating cell events in flow cytometry experiments is presented in Supplementary Fig. 27.

### General microplate reader experiment protocol

Unless otherwise indicated, overnight cultures were diluted 1:200 into fresh M9-Gluc media with their respective inducer concentrations. In AHL-LuxR and Van-VanRAM inducible circuit experiments, Cuminic acid was added to overnight cultures to set the PCN, and AHL or Vanillic acid was added at the beginning of the microplate reader experiment. All experiments were carried out in a BioTek Cytation 5 microplate reader operating the Gen5 software. The plate was cultured for 10–15 h (depending on the experiment) at 37 °C, with orbital shaking at 548 cpm with an orbital diameter of 2 mm. The following measurements were performed every 15 min: optical density at 700 nm, green fluorescence at Ex: 485/10 nm and Em: 515/10 nm with gain 80, and red fluorescence at Ex: 565/15 nm and Em: 600/15 nm with gain 100. All microplate reader experiments had triplicate controls of blank media, NonFP cells, and if appropriate, cells with constitutive mScarI expression from the genome (i.e., DHScarI and BWScarI). Further

detailed data using kinetic microplate reader experiments are presented in Supplementary Section 1.10.

## Chi.Bio turbidostat experiment protocol

Chi.Bio turbidostats[36] pre-assembled by LabMaker were utilized for the experiments presented in Fig. 4a and Supplementary Fig. 9. M9-Gluc media supplemented with appropriate antibiotics and Cuminic acid were used for all Chi.Bio experiments. The Operations Manual (version 1.2) available at https://chi.bio/operation/ was followed for setting up the fluidic lines, electrical connections, user interface (Operating Software v2.3), and for initiating the experiments. At the start of the experiment, each chamber was filled with 20 mL M9-Gluc media, into which the overnight culture was diluted 1:200. The Chi.Bio temperature was set to 37 °C, and the sensors were set to monitor GFP (Ex: 457/35 nm, Em: 510/40 nm, Gain: 512×) and optical density at 600 nm (OD600) at 1 min intervals for 48 h. The target OD600 to maintain was set to 0.5 AU, with an upper and lower bound of 0.6 AU and 0.4 AU, respectively. The media reservoir with appropriate antibiotic and inducer concentrations was changed every 12 h for the duration of the experiment, at the same time 20 μL of cell culture was also sampled and analyzed in the flow cytometer to quantify sfGFP signal distribution across the population.

## Plasmid stability experiment

For the experiments presented in Fig. 4b and Supplementary Fig. 10b, overnight pre-cultures were prepared in M9-Gluc media supplemented with appropriate concentrations of Cuminic acid and ampicillin. At the start of the experiment, all overnight cultures were diluted 1:200 into 800 μL fresh media in deep-well plates with appropriate Cuminic acid conditions, and either with or without ampicillin for selection. The cultures were incubated in the plate shaker at 37 °C, set to 800 rpm. Every 2 h, for a total of 14 h, 20–50 μL of cell culture was sampled for analysis by flow cytometry. To maintain cells in continuous growth, the cell cultures were diluted 1:40 into 800 μL of fresh media with appropriate Cuminic acid and ampicillin conditions at 6 and 10 h.

## Plasmid setpoint change experiment

For the experiment presented in Supplementary Fig. 9a, overnight pre-cultures were prepared in M9-Gluc media supplemented with 0.4 μM Cuminic acid and ampicillin. At the start of the experiment, overnight pre-cultures were diluted 1:80 into 800 μL fresh media with 0.4 μM Cuminic acid and ampicillin, in deep-well plates. Cells were incubated in the plate shaker set to 37 °C and shaking at 800 rpm. Every 2 h, for a total of 12 h, 20–50 μL of cell culture was sampled for flow cytometry analysis. At 4 h, cells were diluted 1:40 into fresh 800 μL of media with a higher concentration of Cuminic acid to trigger PCN upregulation. This second Cuminic acid concentration was maintained for the rest of the experiment. To maintain cells in continuous growth, the cell cultures were diluted 1:40 into 800 μL of fresh media at 8 h with appropriate Cuminic acid conditions.

## Genetic toggle switch experiment

For the experiments presented in Fig. 7, overnight pre-cultures were prepared in M9-Gluc media supplemented with ampicillin, and appropriate Cuminic acid concentrations when the toggle switch was harbored in TULIP; no Cuminic acid was added for strains carrying the toggle switch on plasmids with fixed copy number. At the start of the experiment, overnight pre-cultures were diluted 1:800 into 800 μL of fresh media supplemented with the appropriate concentrations of ampicillin, Cuminic acid, and either 1 mM IPTG or 80 nM aTc. The cells were incubated in a plate shaker set to 37 °C shaking at 800 rpm. At 8 h, 20 μL of cell culture was sampled and analyzed by flow cytometry ("pre-wash" sample). At 12 h post induction with IPTG or aTc, cells were diluted 1:800 into 800 μL of fresh media without IPTG or aTc, and were cultured overnight in the plate shaker. At 24 h, the overnight culture

was diluted 1:800 into 800 μL of fresh media of the same respective media conditions lacking IPTG and aTc, and the culture was incubated for an additional 8 h in the plate shaker. At 32 h of the experiment, 20 μL of cell culture was sampled and analyzed by flow cytometry ("post-wash" sample).

## CRISPRi experiment

For the data presented in Fig. 8 and Supplementary Fig. 26, all experiments were conducted in FR-E01 (Supplementary Table 1) and all guide RNA designs can be found in Supplementary Table 10. Overnight pre-cultures were prepared in M9-Gluc media supplemented with ampicillin to select for TULIP, and kanamycin to select for the gRNA expression plasmid (p15A-kanR vector). A final concentration of 1 μM aTc[43] was used to activate dCas9 expression from the genome, and 0.4 μM Cuminic acid was used to stably maintain TULIP at a basal copy number. At the overnight pre-culture step, appropriate concentrations of Van or AHL were added to induce gRNA expression, which in turn allows TULIP to reach its setpoint copy number over the duration of the overnight pre-culture. For the TULIP positive controls, FR-E01 cells carrying only TULIP were induced with appropriate concentrations of Cuminic acid, and no other inducer. On the day of the experiment, overnight pre-cultures were diluted 1:200 into 200 μL fresh media with appropriate media conditions and were loaded into the microplate reader to monitor growth and sfGFP expression as proxy for PCN.

## Data analysis

For data analysis and plotting the results, Python (v3.8.5) was used in the JupyterLab interface (v2.2.6), provided in the Anaconda package (v1.7.2). Inkscape (v1.0.2) and Adobe Illustrator (v26.5) were used for creating figures. For qPCR experiments, Ct values were extracted from the raw data using the provided software with the instrument (QuantStudio 7 Pro, Design and Analysis v2.5), PCN value calculations were done in Microsoft Excel and visualized in Python. Flow cytometry experiments were analyzed and visualized using Python, first gating events according to forward and side scatter heights to include at least 10,000 cell events (Supplementary Fig. 27). The median fluorescence signal is calculated for each sample to represent the relative gene expression of the population. For the toggle switch experiments, the relative eGFP and mKate2 signals were plotted and gated to categorize events as either Green, Red or NonFP (Supplementary Figs. 16–18). Microplate reader experiments were analyzed and visualized using Python, specific growth rate and fluorescence production rate were calculated for each well as

$$\mu = \frac{\ln(OD_{700}(t_3)) - \ln(OD_{700}(t_1))}{t_3 - t_1}, \qquad FPR = \frac{FP(t_3) - FP(t_1)}{(t_3 - t_1)OD_{700}(t_2)},$$

respectively, where $OD_{700}(t)$ and $FP(t)$ denote the OD and fluorescent measurements at time $t$, and $t_1$, $t_2$, and $t_3$ are subsequent data collection timepoints with 15 min spacing[37]. The corresponding values displayed in each figure represent the maximal values throughout the experiment (during exponential growth, defined within a time-window of 0–10 h), localized using the Python function `scipy.signal.find_peaks()`. Data presented in Supplementary Figs. 28 and 29 illustrate that our findings hold across a variety of timepoints. All mScarI production values were normalized to a respective control strain (i.e., DHScarI or BWScarI) so that the outputs of the sensor modules can be compared across the different strains of *E. coli*. All experiments (including controls) were carried out in biological triplicates, defined as three colonies from each strain to be tested, thus each data point represents a mean of biological triplets, and the error bars represent the corresponding standard deviations. Numerical simulations in Supplementary Section 3 were carried out using MATLAB 2022a.

**Reporting summary**

Further information on research design is available in the Nature Portfolio Reporting Summary linked to this article.

## Data availability

Source Data are provided with this paper.

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

## Acknowledgements

This work was supported by research funds provided to A.G. by New York University Abu Dhabi. We thank the support that we received with flow cytometry experiments from the Center for Genomics and Systems Biology and the Core Technology Platform at NYUAD.

## Author contributions

S.H.-N.J. contributed to the conception and design of the work, and to the acquisition, analysis, and interpretation of data, and has drafted the initial submission and the revision. C.Y. contributed to the acquisition, analysis, and interpretation of data. A.G. contributed to the conception and design of the work, and to the analysis, and interpretation of data, and has drafted the initial submission and the revision. All authors approved the submitted version.

## Competing interests

The authors declare no competing interests.
