## [Peer Review File · Nature Communications]

Reviewers' Comments:

Reviewer #1:

Remarks to the Author:

The paper discussed about a new technique, named TULIP, for tuning gene expression via dynamic control of DNA plasmid copy number.

As it introduces a new tuning dimension in a synthetic gene construct design space, it could be a useful tool to complement existing methods that are mostly at the transcriptional or translational level. TULIP's design is based on the reprogrammed tight autoregulation mechanism of the RepA (repAv7), a plasmid replication initiation factor under control of a cuminic acid-inducible promoter (PCymRC). The results of their efforts resulted in tunable plasmid copy numbers in *E. coli*. The authors demonstrated that TULIP could work across an array of *E. coli* strains for different applications. Subsequently, the authors argued that TULIP may facilitate rapid prototyping and optimization of gene circuits, as it can reduce the needs for exhaustive screening of a large genetic part's library that are heavily dependent on cloning and transformation work. Furthermore, the authors also demonstrated the portability and transferability of modules developed with TULIP, by providing a means to optimize the modules (via model-based analysis) when reused in a new genetic context.

The manuscript is concise and well written. The figures are clear. The authors successfully demonstrate that TULIP could become a useful approach in the synthetic biology toolkit. However, below comments for manuscript improvement need to be addressed.

1. TULIP's hypersensitive response to cuminic acid in protein expression strains (e.g., NEBExpress): perhaps the authors can address this issue by designing an alternative TULIP variant bearing a stronger degradation tag?
2. Versatility of TULIP: It would be nice if the authors could demonstrate that TULIP can be induced and controlled by other inducible promoters as well.
3. Throughout the manuscript, authors have presented 3 relatively simple bio-sensors to support their claim on how TULIP can be employed for gene expression control. While the results are clear, it would be convincing to see TULIP's applicability to work with a more complex circuit, such as some of the well-established architectures, e.g. toggle switch, IFFL circuit, AND/OR gates, etc.
4. For the module's reusability, in Figure 5, authors only demonstrated 2 out of 3 examples from Figure 4, stating that "an implementation is considered successful if in the ON state the output exceeds a pre-defined threshold value". Does it mean authors have attempted to implement the third example yet being unsuccessful? Will TULIP's efficiency (in terms of reusability) decrease as more complexity is added to the circuits to be designed? Discussions on how TULIP can work with more complex scenarios will be useful.
5. Figure 5: at first glance, I did not immediately realize that the red and yellow color coding in the graphs depicted the RBS differences. I think it will be easier for readers if this information is included in the caption and/or label in the plot itself.
6. The discussion of TULIP in resolving synthetic gene circuits known issues such as leaky expression, narrow ranges of activation, repression control, and metabolic burden seems limited. What is the specific advantage here, and what are the next steps if any of these issues are needed for improvement? Two related studies could also be discussed: ref. 23 and Metab Eng. 2022,70:67-78. doi: 10.1016/j.ymben.2022.01.003
7. Ref. 35 and ref. 51 are the same, and is ref. 47 appropriate here? please check.

Reviewer #2:

Remarks to the Author:

Overview:

A challenge in bacterial synthetic biology has been the ability to finely control plasmid copy numbers. Recently, plasmid backbones with a wider range of copy numbers have become available, but these require either the use of specific engineered host strains or multiple plasmids with fixed copy numbers. Development of a robust and tunable all-in-one plasmid copy number system that functions regardless of host strain background is yet to be achieved. In the manuscript "TULIP: A portable, modular DNA plasmid with inducible copy number control" the authors use a systematic approach to build and characterize TULIP. Further, they demonstrate how TULIP can be used for rapid characterization and prototyping of a few simple genetic circuits in different standard laboratory *E. coli* strains. The manuscript overall is well organized with an easy to follow logic and could potentially be of interest to synthetic biologists. However, it is not possible to fully and properly evaluate the overall practicality of the plasmid system as there are some concerns regarding specific properties of TULIP that need to be addressed.

Major comments:

1. The authors state that TULIP realizes three key properties (line 63): (i) flexible and dynamic regulation over time; (ii) portability across multiple strains; and (iii) self-containment allowing easy plug-and-play deployment of genetic circuits. While the latter points are addressed, the first point is not adequately demonstrated. Time course experiments are required to support the claim that TULIP can be dynamically regulated over time. The authors do not show any time course traces although the single production rates calculated for each induction condition in Figures 2-5 are derived from 15 hour time course experiments using a microplate reader.
 - a. It is unlikely that the fluorescence production and growth rates for each induction condition were constant for the duration of the experiment, especially considering that cells diluted 1:200 from overnight cultures and grown for 15 hours would likely transition through lag, log, and stationary phase during this time. How were the production rates and growth rates calculated? For example, were all the production rates during the entire time course averaged together or were the production rates during only a specific window of time or growth phase used? This needs to be fully described in the Methods section.
 - b. Regardless of the method of production rate calculation, the full time course traces need to be included in the manuscript. This is necessary to demonstrate that once the cells reach a desired steady-state plasmid copy number they stably maintain the copy number over time. These time courses will also provide information on the minimum time required for experiments based on how long it takes cells to reach the different plasmid copy number setpoints.
 - c. The claim of 'dynamic' regulation over time would also be strengthened if the authors can show that multiple setpoint changes can be achieved with the same cell population over time.
2. A challenge faced when building a plasmid copy number control system is the random nature of plasmid segregation during cell division, which over time can lead to undesirable outcomes ranging from a complete loss of plasmid to a toxic build-up of plasmid in cells. As described in the Discussion and Figures S5 and S6, in the absence as well as lower concentrations of Cuminic acid, at least a portion of the cell population appears to lose the plasmid. The authors suggest this feature could be a benefit in terms of biocontainment and applications requiring conditional or transient plasmid maintenance. However, for routine synthetic biology use, plasmid instability can often be detrimental. It is important that the authors provide data for plasmid stability over time at different plasmid copy numbers with at least one strain. A standard cell viability test or flow cytometry to quantitate any appearance/accumulation of non-fluorescent cells over time should address this, especially for lower plasmid copy number setpoints. If possible, plasmid stability should be examined with and without antibiotics as presence of antibiotics forces selection of cells with plasmid but can also enrich for escapees with upregulated copy numbers (despite the use of negative feedback) and absence of antibiotics allows one to calculate how many cells actually lose plasmid over time if not selected for. If plasmid loss over time is an issue, the authors should include an approximate range of time/generations in which the plasmid can be safely used. Another possibility would be to add a partitioning system to TULIP to ensure that each daughter cell inherits at least one plasmid.
3. The empty TULIP backbone does not seem to significantly alter cell growth rates at the different copy numbers (Figures 2 and 3). However, as shown in Figures 4, S7 and S8, incorporating additional genetic circuits into TULIP can introduce resource burden and affect cell growth. Resource burden and changes in cell growth have global effects (gene expression, dilution rates, etc.), which could also affect the response of TULIP to Cuminic acid. The authors should address

whether or not TULIP copy numbers are robust to these perturbations.

4. The use of the term 'prototyping' should be clarified in the manuscript. Prototyping suggests that TULIP is used to rapidly identify a set of parts and conditions (inducer concentration, plasmid copy number, etc.) that can achieve a desired output such as in Figure 5 but that this prototype is not the final version of the plasmid to be used in downstream applications. Is the TULIP-based construct intended to be the final working plasmid or is the optimized circuit expected to be moved to a standard plasmid backbone? If the latter is the case, to better validate the feasibility of TULIP for prototyping, especially when the circuit is functional only within a narrow plasmid copy number range (Figure 5B), the authors should move the circuits to corresponding standard fixed-copy backbones and show that the circuits perform/do not perform as predicted.

Minor comments:

- All experiments with TULIP are performed in M9 medium. Is this medium necessary for proper TULIP function or can other media be used?
- Line 103: The wording is slightly confusing. It sounds like amino acid substitutions can be made on the pSC101 origin to modulate the dissociation constant between RepA and the pSC101 ori.
- Figure 1D caption: It would be helpful to mention in the caption that the second plasmid also contains wild-type repA under tight autoinhibition.
- Figure 2A: The lightened areas of the figure are difficult to see and should be slightly darker.
- Figure 2A caption: Cuminic acid is misspelled.
- Figure S3: The gates used for each round of screening should be included on the histograms.
- Figure S4: What was the metric used when selecting the ten variants?
- Cell growth is reported as normalized growth rate. The non-normalized growth rates should be included in the Supplement, namely when comparing TULIP in the different host strains.
- Line 191: The proteins should be written as DnaA, DnaB and DnaG.
- OriC is used in this manuscript to denote the pSC101 origin of replication. Use of this label can be somewhat confusing as OriC tends to be specifically used for the chromosomal origin of replication.
- Lines 282 and 290: The modeling section should be SI Section S3.
- Supplementary Section S3: The parameter values used should be justified with either references and/or a brief discussion within this section.

Response to Referees

We thank the Editor and the Reviewers for their constructive feedback that enabled us to improve our work and significantly expand the scope of the manuscript. Minor revisions center around experiments already featured in the original submission:

1. we demonstrate correct functioning of all three sensor modules (originally featured in Figure 4) in both DH10B and BW25113 (Figure 6 and Supplementary Fig. 14);
2. we include the gates and metric used during the screening process of TULIP (Supplementary Fig. 3) and time course trace data for all microplate reader experiments (Supplementary Section 1.10);
3. we verify that TULIP works across a variety of commonly used growth media (Supplementary Fig. 7);
4. we further characterize the behavior of TULIP in NEBExpress both with the original design and also with a stronger degradation tag to demonstrate fine-grain plasmid copy number (PCN) control (Supplementary Fig. 8);
5. we characterize how the PCN of TULIP is affected by metabolic burden (Supplementary Fig. 13).

Major revisions represent three entirely new lines of inquiry in the revised manuscript to (i) further characterize the performance of TULIP, (ii) illustrate its practical applicability, and (iii) increase its versatility, each presented in a separate subsection within the “Results” section:

1. we performed additional experiments to demonstrate dynamic regulation over time with multiple setpoints to verify and characterize plasmid stability (detailed in the new section titled “PCN can be dynamically regulated and reliably maintained using TULIP”);
2. we characterized the behavior of a genetic toggle switch to illustrate that TULIP can be leveraged to facilitate the prototyping of complex genetic circuits and also to demonstrate that modules behave similarly in TULIP and in plasmids with fixed copy number (detailed in the new section titled “TULIP can facilitate gene circuit prototyping”);
3. we extended the original design of TULIP with an additional and easily customizable CRISPRi-based layer of control for increased versatility (detailed in the new section titled “TULIP enables versatile PCN control using a variety of input stimuli”).

Here, we first outline how we revised the manuscript regarding the above three new lines of inquiry, then turn to Reviewer comments (highlighted in blue) and detail how we addressed them one-by-one. In addition to the revised manuscript, we also uploaded a copy indicating all the changes (generated using the `latexdiff` package: added text is blue, discarded text is red).

1) PCN can be dynamically regulated and reliably maintained using TULIP

By combining the Chi.Bio turbidostat platform (1) with flow cytometry analysis, we first demonstrate that PCN can be dynamically adjusted and reliably maintained by varying the Cuminic acid concentration of the growth media, considering both low-to-high and high-to-low transitions over a wide range (Figure 4a and Supplementary Fig. 9a). Second, we characterize that it takes approximately 3–4 hours for cells to reach the different PCN setpoints when using TULIP (Figure 4a, Supplementary Fig. 10a and Supplementary Fig. 11). Third, we confirm that TULIP can be reliably maintained at the same level for extended periods of time (over 50 generations) at both low and high PCN with negligible temporal variation in the presence of antibiotic selection pressure (Supplementary Fig. 9bc). These results verify that over timescales typical in synthetic biology applications, TULIP can be deployed without encountering typical challenges related to plasmid maintenance (e.g., loss of the plasmid from the cell population, or runaway replication leading to toxic build-up), despite the random nature of plasmid segregation during cell division. Loss of TULIP is only observed in the absence of antibiotic selection pressure, a phenomenon that is particularly pronounced at low Cuminic acid concentration (Figure 4b, Supplementary Fig. 10b and Supplementary Fig. 12).

2) TULIP can facilitate gene circuit prototyping

To illustrate that well-established and commonly used complex circuits also function as expected in TULIP, and to highlight that TULIP can facilitate gene circuit prototyping, we first cloned the genetic toggle switch from (2) into TULIP and verified that the module exhibits bistable behavior across a range of PCNs tunable with TULIP (Figure 7). We also verified that data obtained using TULIP accurately predict the behavior observed when relying on plasmids with matching fixed copy number harboring the same toggle switch. This demonstrates that TULIP can facilitate gene circuit prototyping via PCN control (e.g., observed steady states of the toggle switch span approximately a 20-fold range) not only when the developed module is deployed in TULIP itself, but also when it is moved to plasmid backbones with a fixed copy number.

3) TULIP enables versatile PCN control using a variety of input stimuli

In the revised manuscript we included a second CRISPRi-based layer of control interfacing with TULIP to significantly extend its applicability. By recruiting dCas9, in this expanded scheme a guide RNA targets and inhibits CymRAM expression, offering an alternative mode of control for PCN regulation (complementing Cuminic acid). The expression of this guide RNA can be coupled to a wide array of input stimuli (e.g., chemical inducers, stress, pH, temperature) to enable tunable PCN control, similar to induction with Cuminic acid, as we illustrate in Figure 8 using Van and AHL without any modifications to TULIP. Thus, TULIP together with this flexible and versatile additional layer of control provides a powerful way of interrogating and tuning gene circuit behavior in diverse contexts.

Reviewer #1

The paper discussed about a new technique, named TULIP, for tuning gene expression via dynamic control of DNA plasmid copy number. As it introduces a new tuning dimension in a synthetic gene construct design space, it could be a useful tool to complement existing methods that are mostly at the transcriptional or translational level. TULIP's design is based on the reprogrammed tight autoregulation mechanism of the RepA (repAv7), a plasmid replication initiation factor under control of a cuminic acid-inducible promoter (PCymRC). The results of their efforts resulted in tunable plasmid copy numbers in *E. coli*. The authors demonstrated that TULIP could work across an array of *E. coli* strains for different applications. Subsequently, the authors argued that TULIP may facilitate rapid prototyping and optimization of gene circuits, as it can reduce the needs for exhaustive screening of a large genetic part's library that are heavily dependent on cloning and transformation work. Furthermore, the authors also demonstrated the portability and transferability of modules developed with TULIP, by providing a means to optimize the modules (via model-based analysis) when reused in a new genetic context. The manuscript is concise and well written. The figures are clear. The authors successfully demonstrate that TULIP could become a useful approach in the synthetic biology toolkit. However, below comments for manuscript improvement need to be addressed.

We thank the Reviewer for their constructive feedback and suggestions that helped us not only improve the results presented in the original submission, but also sparked novel lines of research inquiry significantly expanding the scope of our work. For instance, the revised manuscript now features (i) the toggle switch to illustrate TULIP's applicability to work with more complex circuits; and (ii) an expanded control scheme interfacing TULIP for increased versatility via a CRISPRi-based regulatory layer to enable PCN modulation in response to a wide range of input stimuli. In what follows we address the helpful comments that we have received one-by-one.

1. TULIP's hypersensitive response to cuminic acid in protein expression strains (e.g., NEBExpress): perhaps the authors can address this issue by designing an alternative TULIP variant bearing a stronger degradation tag?

We thank the reviewer for this invaluable suggestion. The response of TULIP to Cuminic acid in NEBExpress appears considerably less gradual than in the other strains when using the same inducer concentration gradient, however, this issue can be easily mitigated by considering a finer concentration gradient of Cuminic acid (Supplementary Fig. 8a). This phenomenon likely stems from the lack of Lon and OmpT protease machinery in NEBExpress (3), contributing to a lower removal rate of RepAv7, thus increasing sensitivity of PCN to lower levels of Cuminic acid when compared to the other strains. To increase the removal rate of RepAv7, we followed the Reviewer's suggestion and modified TULIP by replacing the moderate AAV degradation tag with a stronger LVA tag (4) (Supplementary Fig. 8b). This modification lowered the minimum PCN that can be stably maintained, improving the dynamic range of TULIP in NEBExpress from 15-fold to 32-fold (Supplementary Fig. 8c), however, TULIP's response to Cuminic acid induction appears nearly identical with the two degradation tags. This suggests that should further improvements be deemed necessary, simply using an even stronger degradation tag may not be sufficient; instead, a process similar to the one outlined in Supplementary Section 1.2 would need to be repeated.

2. Versatility of TULIP: It would be nice if the authors could demonstrate that TULIP can be induced and controlled by other inducible promoters as well.

TULIP was designed with modularity and orthogonality in mind to enable its adoption with ease and minimal disruption to already existing systems. By using Cuminic acid together with the PCymRC promoter as the primary mode of PCN control (thus avoiding staple chemical inducers such as IPTG and aTc), TULIP is cross-compatible with a wide array of commonly used genetic components. While replacing the PCymRC promoter in TULIP with other variants would provide us with PCN control responsive to inducers other than Cuminic acid, for each variant a process similar to the one outlined in Supplementary Section 1.2 would need to be repeated, which would represent an arduous, time-consuming challenge.

Alternatively, in the revised manuscript we included a second layer of control interfacing with TULIP to significantly extend its applicability (Figure 8). In this expanded scheme a guide RNA targets and inhibits CymRAM expression by recruiting dCas9, thus upregulating the expression of this guide RNA has an effect similar to Cuminic acid induction. By relying on the pSC101 origin from incompatibility group C as our scaffold (thus avoiding the most commonly used plasmids in incompatibility group A and B), TULIP is cross-compatible with a wide array of commonly used vectors that can be deployed to harbor this secondary layer of control. Thus, in this expanded architecture PCN control via TULIP offers a versatile and powerful new way of interrogating and tuning gene circuit behavior in response to a wide array of input stimuli (e.g., chemical inducers, stress, pH, temperature). We demonstrate this in Figure 8 using Van and AHL (without any modifications to TULIP) resulting in PCN control similar to what we originally obtained via modulating Cuminic acid concentration. While in the proof-of-concept implementation of Figure 8 we rely on FR-E01 for dCas9 expression, this gene can also be integrated into the plasmid harboring the additional layer of control to ensure multi-strain portability even of this expanded control scheme.

3. Throughout the manuscript, authors have presented 3 relatively simple bio-sensors to support their claim on how TULIP can be employed for gene expression control. While the results are clear, it would be convincing to see TULIP's applicability to work with a more complex circuit, such as some of the well-established architectures, e.g. toggle switch, IFFL circuit, AND/OR gates, etc.

Following this recommendation, we cloned the genetic toggle switch from (2) into both TULIP and plasmids with fixed copy number. In both of these implementations, we verify that the module exhibits bistable behavior across a range of PCNs and we demonstrate that we obtain consistent behavior across the two types of backbones (tunable and fixed). This illustrates that similarly to the sensor modules originally featured in the manuscript, more complex genetic circuits function as expected when cloned into TULIP, and highlights that TULIP can facilitate prototyping via flexible and dynamic PCN control (e.g., observed steady states of the toggle switch span a 20-fold range at varying levels of Cuminic acid induction) not only when the developed module is deployed in TULIP itself, but also when it is moved to plasmids with fixed copy number.

4. For the module's reusability, in Figure 5, authors only demonstrated 2 out of 3 examples from Figure 4, stating that "an implementation is considered successful if in the ON state the output exceeds a pre-defined threshold value." Does it mean authors have attempted to implement the third example yet being unsuccessful? Will TULIP's efficiency (in terms of reusability) decrease

as more complexity is added to the circuits to be designed? Discussions on how TULIP can work with more complex scenarios will be useful.

We thank the Reviewer for highlighting a potentially misleading interpretation of the results (i.e., that the NOT gate could not be successfully re-used in BW25113). To address this issue, we have now included the behavior of all three modules in both *E. coli* strains (DH10B and BW25113). The data in Figure 6 for the single-layer sensor modules and in Supplementary Fig. 14 for the NOT gate demonstrate that all three circuits display the expected input-output behavior in both DH10B and BW25113 at all tested levels of PCN. Additionally, we have also demonstrated in Figure 7 that the toggle switch from (2) behaves similarly in TULIP and in plasmids with matching fixed copy number. Taken together, these results suggest that TULIP can be successfully deployed as a backbone for genetic circuits typically used in synthetic biology. In terms of the re-usability of modules, while we expect TULIP to offer a valuable dimension for tuning the behavior of genetic circuits irrespective of their size, the beneficial impact of flexible PCN control will likely have to be complemented with other regulatory and cloning strategies in case of increasing circuit complexity due to the quickly growing number of strain-dependent biophysical parameters (this is now explicitly indicated in the section titled “TULIP promotes the re-use of modules in different contexts”).

5. Figure 5: at first glance, I did not immediately realize that the red and yellow color coding in the graphs depicted the RBS differences. I think it will be easier for readers if this information is included in the caption and/or label in the plot itself.

Following this helpful suggestion, we have (i) revised the figure to clearly indicate the relative strength of the RBSs and the corresponding color scheme; and (ii) explicitly included this information in the caption of Figure 6 as well (featured originally as Figure 5).

6. The discussion of TULIP in resolving synthetic gene circuits known issues such as leaky expression, narrow ranges of activation, repression control, and metabolic burden seems limited. What is the specific advantage here, and what are the next steps if any of these issues are needed for improvement? Two related studies could also be discussed: ref. 23 and *Metab Eng.* 2022,70:67-78. doi: 10.1016/j.ymben.2022.01.003

We thank the Reviewer for pointing out this limitation and the helpful guidance on how to address this shortcoming. The Discussion has been revised accordingly. We include the works of Li *et al.* (2022) and Roches *et al.* (2022) and explain how TULIP contributes to the recent and growing body of tools which aim to enable PCN control as an accessible parameter for biological engineering. We highlight that TULIP was designed with modularity and orthogonality in mind to enable its adoption with ease and minimal disruption to already existing systems. By using Cumenic acid as the primary mode of PCN control and the pSC101 origin from incompatibility group C as our scaffold (thus avoiding staple chemical inducers such as IPTG and aTc, and plasmids in incompatibility group A and B such as pColE1, pBR322, pUC19, and p15A), TULIP is cross-compatible with a wide array of commonly used vectors and genetic components. Combining this with the fact that TULIP has demonstrated robust functionality considering a range of *E. coli* hosts, media compositions, and synthetic gene circuits, it offers a flexible platform that can be deployed effortlessly in vastly different contexts to mitigate known issues of synthetic gene circuits (such

as leaky expression, narrow ranges of activation, repression control, and metabolic burden), which we highlight using our own data featured in the paper and also by referencing published results.

7. Ref. 35 and ref. 51 are the same, and is ref. 47 appropriate here? please check.

We thank the Reviewer for bringing this issue to our attention. References 35 and 51 were indeed duplicates, this is now corrected (appearing now as reference 38). Reference 47 outlines the electrocompetent cell preparation protocol used in our study, hence it is cited in the Methods section both in the original and in the revised manuscript (appearing now as reference 71).

Reviewer #2

Overview: A challenge in bacterial synthetic biology has been the ability to finely control plasmid copy numbers. Recently, plasmid backbones with a wider range of copy numbers have become available, but these require either the use of specific engineered host strains or multiple plasmids with fixed copy numbers. Development of a robust and tunable all-in-one plasmid copy number system that functions regardless of host strain background is yet to be achieved. In the manuscript “TULIP: A portable, modular DNA plasmid with inducible copy number control” the authors use a systematic approach to build and characterize TULIP. Further, they demonstrate how TULIP can be used for rapid characterization and prototyping of a few simple genetic circuits in different standard laboratory *E. coli* strains.

The manuscript overall is well organized with an easy to follow logic and could potentially be of interest to synthetic biologists. However, it is not possible to fully and properly evaluate the overall practicality of the plasmid system as there are some concerns regarding specific properties of TULIP that need to be addressed.

We thank the Reviewer for their constructive feedback and suggestions. These have helped us not only improve the results presented in the original submission, but also sparked novel lines of research inquiry significantly expanding the scope of our work. For instance, in the revised manuscript we (i) demonstrate dynamic PCN regulation through multiple setpoint changes with the same cell population over time; (ii) characterize the time that is required for cells to reach the different PCN setpoints; (iii) quantify plasmid stability both in the presence and absence of antibiotic selection pressure; (iv) verify PCN control using TULIP across commonly used growth media; (v) characterize the effects of perturbations in growth rate and metabolic burden on PCN; and (vi) demonstrate that TULIP can facilitate gene circuit prototyping not only when the modules are deployed in TULIP but also if they are cloned into plasmids with matching fixed copy number. In what follows we address the helpful comments that we have received one-by-one.

Major comments:

1. The authors state that TULIP realizes three key properties (line 63): (i) flexible and dynamic regulation over time; (ii) portability across multiple strains; and (iii) self-containment allowing easy plug-and-play deployment of genetic circuits. While the latter points are addressed, the first point is not adequately demonstrated. Time course experiments are required to support the claim that TULIP can be dynamically regulated over time. The authors do not show any time course traces although the single production rates calculated for each induction condition in Figures 2-5 are derived from 15 hour time course experiments using a microplate reader.

We thank the Reviewer for pointing out this shortcoming of our manuscript. Below we detail how we addressed this issue both by presenting data already collected and by performing additional experiments not only using flow cytometry analysis, but also leveraging the Chi.Bio turbidostat platform (*I*) to monitor the performance of TULIP for approximately 50 generations.

- (a) It is unlikely that the fluorescence production and growth rates for each induction condition were constant for the duration of the experiment, especially considering that cells diluted 1:200 from overnight cultures and grown for 15 hours would likely transition through lag, log, and stationary phase during this time. How were the production rates and growth rates calculated? For example, were all the production rates during the entire time course averaged together or were the production rates during only a specific window of time or growth phase used? This needs to be fully described in the Methods section.

We thank the Reviewer for pointing out this issue. Cells indeed transition through the above phases, leading to temporal variations in both fluorescence production and growth rate (kinetic microplate reader data are now presented for all experiments in Supplementary Section 1.10). The formulas for growth rate and fluorescent production rate calculations are included in the Methods section (under “Data analysis”), and the corresponding values displayed in each figure represent the maximal values throughout the experiment, localized using the Python function `scipy.signal.findpeaks()`, now explicitly stated in the Methods (under “Data analysis”). Notably, data presented in Supplementary Fig. 28–29 illustrate that our findings hold across a variety of timepoints. In Supplementary Fig. 28 we replicate the analysis presented in Figure 3d to demonstrate that TULIP’s response to Cuminic acid in all tested strains is qualitatively similar considering sfGFP production rate at its peak or its value when growth rate is maximized. Similarly, data in Supplementary Fig. 29 illustrate that metabolic burden due to mScarI expression has a similar impact on sfGFP production considering the sensor modules in Figure 5 at different timepoints.

- (b) Regardless of the method of production rate calculation, the full time course traces need to be included in the manuscript. This is necessary to demonstrate that once the cells reach a desired steady-state plasmid copy number they stably maintain the copy number over time. These time courses will also provide information on the minimum time required for experiments based on how long it takes cells to reach the different plasmid copy number setpoints. We have included the kinetic microplate reader data for all experiments in Supplementary Section 1.10. As cells transition through lag, log, and stationary phase during these experiments, fluorescence production shows temporal variation. To demonstrate that once cells reach a desired steady-state PCN they stably maintain it over time, we performed additional experiments using both the Chi.Bio turbidostat platform (1) and further flow cytometry analysis to monitor both population and single-cell level cellular behavior.

First, DH10B cells harboring the empty TULIP backbone were grown for 48 hours using the Chi.Bio platform relying on its turbidostat functionality. According to the data in Supplementary Fig. 9bc, optical density measured at 600 nm wavelength of light (OD600) is maintained at the pre-specified constant level of 0.5 AU (± 0.1 AU) to ensure that cells remain in exponential growth phase throughout the experiment. Growth media contained either 0.4 μM or 5.0 μM Cuminic acid, corresponding to low and high PCN, respectively. The results in Supplementary Fig. 9bc confirm that TULIP’s PCN can be reliably maintained for extended periods of time, over 50 generations in each case. Plasmid stability is further confirmed via flow cytometry analysis both when Cuminic acid is modulated (Supplementary Fig. 10a and Supplementary Fig. 11) and when it is kept constant (Supplementary Fig. 10b and Supplementary Fig. 12).

Additionally, the data presented in Supplementary Fig. 10a and Supplementary Fig. 11 reveal that it takes approximately 3–4 hours for cells to reach the different PCN setpoints subsequent to instantaneous changes in Cumenic acid concentration. This is further confirmed in the Chi.Bio experiment presented in Figure 4a, exposing that the typical timescale required for cells to reach the different PCN setpoints corresponds to approximately 3–5 generations considering that the specific growth rate is approximately $1\text{--}1.2\text{h}^{-1}$ (independent of PCN).

- (c) The claim of ‘dynamic’ regulation over time would also be strengthened if the authors can show that multiple setpoint changes can be achieved with the same cell population over time.

We thank the Reviewer for this invaluable suggestion. To strengthen this claim, DH10B cells harboring the empty TULIP backbone were grown for 48 hours using the Chi.Bio turbidostat platform (1). According to the data in Supplementary Fig. 9a, optical density measured at 600 nm wavelength of light (OD600) is maintained at a pre-specified constant level of 0.5 AU (± 0.1 AU) to ensure that cells remain in exponential growth phase throughout the experiment. Fresh growth media in subsequent phases (lasting 12 hours each) contained varying levels of Cumenic acid to modulate PCN. The fluorescence data in Figure 4a reveal that PCN can be dynamically adjusted and reliably maintained (over 10 generations) considering both low-to-high and high-to-low transitions over a wide range, without noticeable effect on growth rate. Further single cell level analysis of samples taken prior to each transition confirm population-level uniformity (Figure 4a). Taken together, these results demonstrate that PCN can be dynamically regulated and that multiple setpoint changes can be achieved with the same cell population over time.

2. A challenge faced when building a plasmid copy number control system is the random nature of plasmid segregation during cell division, which over time can lead to undesirable outcomes ranging from a complete loss of plasmid to a toxic build-up of plasmid in cells. As described in the Discussion and Figures S5 and S6, in the absence as well as lower concentrations of Cumenic acid, at least a portion of the cell population appears to lose the plasmid. The authors suggest this feature could be a benefit in terms of biocontainment and applications requiring conditional or transient plasmid maintenance. However, for routine synthetic biology use, plasmid instability can often be detrimental. It is important that the authors provide data for plasmid stability over time at different plasmid copy numbers with at least one strain. A standard cell viability test or flow cytometry to quantitate any appearance/accumulation of non-fluorescent cells over time should address this, especially for lower plasmid copy number setpoints. If possible, plasmid stability should be examined with and without antibiotics as presence of antibiotics forces selection of cells with plasmid but can also enrich for escapees with upregulated copy numbers (despite the use of negative feedback) and absence of antibiotics allows one to calculate how many cells actually lose plasmid over time if not selected for. If plasmid loss over time is an issue, the authors should include an approximate range of time/generations in which the plasmid can be safely used. Another possibility would be to add a partitioning system to TULIP to ensure that each daughter cell inherits at least one plasmid.

We thank the Reviewer for pointing out this potential issue. The population-level and single-cell analysis we detail above (when addressing points 1a, 1b, and 1c) demonstrate that TULIP is reliably maintained over multiple generations both at low and high PCN values with negligible

temporal variation. These results verify that over timescales typical in synthetic biology applications, TULIP can be deployed without encountering the above mentioned undesirable outcomes and plasmid maintenance challenges (ranging from complete loss to a toxic build-up in cells), despite the random nature of plasmid segregation during cell division.

Additionally, we performed further flow cytometry experiments to characterize plasmid stability, both in the presence and absence of antibiotic selection pressure. Following the recommendation of the Reviewer, we grew DH10B cells harboring the empty TULIP backbone at various induction levels of Cuminic acid for over 10 generations, or approximately 14 hours of sustained continuous growth, which represents the typical time scale for synthetic biology culture experiments (i.e., overnight cultures, plate reader experiments, etc.). Data presented in Figure 4b, Supplementary Fig. 10b and Supplementary Fig. 12 confirm that while TULIP is reliably maintained at all PCN levels in the presence of antibiotic selection pressure, in its absence TULIP is lost gradually over time. This phenomenon is particularly pronounced at low Cuminic acid concentrations, otherwise cells tend to maintain TULIP even in the absence of antibiotic selection pressure as plasmid half-life across the population well exceeds 50 hours.

3. The empty TULIP backbone does not seem to significantly alter cell growth rates at the different copy numbers (Figures 2 and 3). However, as shown in Figures 4, S7 and S8, incorporating additional genetic circuits into TULIP can introduce resource burden and affect cell growth. Resource burden and changes in cell growth have global effects (gene expression, dilution rates, etc.), which could also affect the response of TULIP to Cuminic acid. The authors should address whether or not TULIP copy numbers are robust to these perturbations.

We thank the Reviewer for pointing out this shortcoming of the original submission. To address it, we performed additional experiments to probe the effects of both metabolic burden and growth rate on PCN when using TULIP.

To reveal the impact of metabolic burden, we selected the sensor module from Figure 5b as an illustrative example as Van-induced mScarI expression leads to approximately 75% reduction in sfGFP expression (Figure 5b) at the highest tested Cuminic acid concentration (2.4 μM , the indicated 4.0 μM level in Figure S7 of the original SI was a typo that has been corrected). According to the qPCR data presented in Supplementary Fig. 13 considering both TULIP and plasmids with fixed copy number, this increased metabolic burden has minor impact on the PCN of either type of vectors despite the significant reduction in sfGFP expression. In case of TULIP, the only potential exception may be at 0.8 μM Cuminic acid concentration as PCN is estimated to decrease from approximately 5 to 3 upon Van induction. However, a similar change can be observed in the opposite direction in case of p15A (from approximately 5 to 10), suggesting that these fluctuations at low PCN levels may be independent of Van induction. Furthermore, comparing the performance of TULIP and plasmids with fixed copy number from the perspective of PCN (Supplementary Fig. 13), growth rate (Supplementary Fig. 35), and mScarI production (Supplementary Fig. 13), the results indicate that metabolic burden has comparable impact on these two types of vectors. This finding is further strengthened by the appearance of the “isocost lines” in Figure 5 (considering two different modules), linking the expression of two unrelated proteins due to competition for shared resources, originally observed when using plasmids with fixed copy number.

To characterize the effects of more global influences on PCN, such as growth rate and nutritional

availability, we tested the behavior of the empty TULIP backbone in four commonly used growth media: M9 supplemented with glucose (M9-Gluc), M9 supplemented with glycerol (M9-Gly), Lysogeny Broth (LB), and Super Optimal Broth (SOB). The data presented in Supplementary Fig. 7 confirm that the induction of TULIP has negligible impact on growth rate across all tested media and Cuminic acid concentration. More importantly, while cells in M9-Gly grew significantly slower than those in M9-Gluc (considering the approximately 33% reduction in growth rate from $\sim 1.2\text{h}^{-1}$ to $\sim 0.8\text{h}^{-1}$), the measured controllable PCN range to various levels of Cuminic acid induction is almost identical (2–48 PCN in M9-Gluc and 3–47 in M9-Gly). While growth rate differences in M9-Gluc and M9-Gly seem to have negligible impact on PCN, they likely represent a major contributor to the observed differences in gene expression: e.g., at the highest tested concentration ($5.0\ \mu\text{M}$ Cuma) flow cytometry analysis reveals that sfGFP expression in M9-Gluc is approximately 2.2-fold higher than in M9-Gly, despite the nearly identical PCN values (Supplementary Fig. 7).

4. The use of the term ‘prototyping’ should be clarified in the manuscript. Prototyping suggests that TULIP is used to rapidly identify a set of parts and conditions (inducer concentration, plasmid copy number, etc.) that can achieve a desired output such as in Figure 5 but that this prototype is not the final version of the plasmid to be used in downstream applications. Is the TULIP-based construct intended to be the final working plasmid or is the optimized circuit expected to be moved to a standard plasmid backbone? If the latter is the case, to better validate the feasibility of TULIP for prototyping, especially when the circuit is functional only within a narrow plasmid copy number range (Figure 5B), the authors should move the circuits to corresponding standard fixed-copy backbones and show that the circuits perform/do not perform as predicted.

We thank the Reviewer for this invaluable comment about “prototyping” and the need for further clarification. In the revised manuscript we removed the term “prototyping” from the sections focusing on the tuning and re-using of the sensor modules harbored in TULIP (results featured in our original submission). Instead, we included a new section titled “TULIP can facilitate gene circuit prototyping” to illustrate how modules created and developed in TULIP can be cloned into plasmids with fixed copy number for deployment in further downstream applications.

To illustrate how TULIP can facilitate prototyping in this sense, we focused on the toggle switch from (2), underpinned by considerably richer dynamics than the sensor modules in Figures 5–6, with the potential for bistability. We consider the standard realization of the toggle switch (5) with LacI (co-expressed with mKate2) and TetR (co-expressed with eGFP) repressing each other (Figure 7a). Therefore, addition of IPTG/aTc results in upregulation of eGFP/mKate2 and downregulation of mKate2/eGFP, respectively. To ensure that the steady state concentration of LacI/TetR match the input dynamic range of further downstream modules, the protein expression levels can be modified via the genetic parts themselves (e.g., by adjusting promoter and RBS strengths), however, this combinatorial library-based approach requires extensive cloning and transformation. Alternatively, using TULIP we may tune them by controlling the PCN via the addition of Cuminic acid without modifying the genetic layout, and once a PCN with desired circuit dynamics has been identified, clone the construct into the appropriate plasmid with matching fixed copy number.

To illustrate this, the genetic toggle switch in Figure 7a was cloned into TULIP and into plasmids with fixed copy number spanning the same PCN range in DH10B. After overnight growth of 12

hours, cells were induced with either IPTG or aTc, then the first round of samples were collected for flow cytometry analysis after 8 hours (pre-wash). As expected, cells induced with IPTG/aTc had elevated levels of eGFP/mKate2 expression, and this shift increased with PCN. To test stability of these induced steady states (and thus the bistability of the toggle switch at different PCN), following 4 more hours of growth cells were diluted and grown for 12 hours in fresh media lacking both IPTG and aTc, then diluted again into fresh media before collecting the second round of samples for flow cytometry analysis after 8 hours (post-wash).

From a qualitative perspective, the behavior of the toggle switch when harbored in plasmids with fixed copy number closely matches that observed when using TULIP, as eGFP and mKate2 expression (both pre-wash and post-wash) follow the same trend independent of the backbone (Figure 7c). Furthermore, while pre-wash and post-wash samples are nearly identical at medium and high PCN levels, there is significant decrease of eGFP when PCN becomes sufficiently low, yielding almost identical post-wash concentrations subsequent to IPTG and aTc induction in case of both types of backbones. In addition to these qualitative similarities, data obtained using TULIP accurately predict the behavior observed when relying on plasmids with matching fixed copy number (Figure 7d) from a quantitative perspective as well (using sfGFP expression from the empty backbone as a proxy for PCN). Finally, as the eGFP decrease in post-wash samples at low PCN values subsequent to IPTG induction is likely a result of low basal expression of LacI from the chromosome in the original DH10B strain, we repeated the experiments with this gene knocked out from the genome. As expected, this modification had identical impact at low PCN levels (reduced drop in post-wash eGFP expression subsequent to IPTG induction), and more importantly, preserved the close quantitative alignment between the behavior observed when using the two types of backbones (Supplementary Fig. 20).

These findings demonstrate not only that PCN control offers a simple and convenient way to tune circuit dynamics (e.g., observed steady states of the toggle switch span approximately a 20-fold range), but also that the type of the backbone (TULIP or plasmids with fixed copy number) has minimal impact on the observed behavior. Thus, TULIP can facilitate gene circuit prototyping via flexible and dynamic adjustment of PCN without relying on time-consuming and error-prone cloning and transformation steps to guide the selection of parts and conditions (e.g., inducer concentration, PCN range) to ensure correct functioning (e.g., desired stability profile, dynamic range) even when modules are eventually deployed in plasmids with matching fixed copy number. Importantly, however, while TULIP provides us with a powerful dimension for tuning circuit behavior, it complements rather than replaces tuning via combinatorial libraries (6). For instance, a toggle switch relying either on monomeric repressors or on unbalanced production rate constants may not be rendered bistable by only modifying the PCN of the plasmid harboring it (Supplementary Fig. 106).

Minor comments:

- All experiments with TULIP are performed in M9 medium. Is this medium necessary for proper TULIP function or can other media be used?

M9 media was originally selected for all experiments due to its significantly lower autofluores-

cence relative to rich medias, thus offering a critical advantage in fluorescence-based characterization experiments.

However, by performing additional qPCR, flow cytometry, and kinetic microplate reader experiments (Supplementary Fig. 7), we verified that TULIP exhibits inducible PCN control in a variety of growth medias typically used in experimental microbiology (M9-Glucose, M9-Glycerol, Lysogeny Broth, and Super Optimal Broth), which is now explicitly stated in the manuscript. As expected, however, media composition can have considerable impact on the PCN ranges: for instance, those observed in Lysogeny Broth and Super Optimal Broth significantly exceed those in M9-Glucose and M9-Glycerol (PCN range is approximately 1–50 and 20–200 in controlled and rich medias, respectively).

- Line 103: The wording is slightly confusing. It sounds like amino acid substitutions can be made on the pSC101 origin to modulate the dissociation constant between RepA and the pSC101 ori.

This sentence has been revised to increase clarity and now reads as “*Introducing mutations at the RepA dimerization interface (e.g., via amino acid substitution) alters the binding kinetics of the RepA monomers and modulates the dissociation constant, thus resulting in mutant variants equipped with different fixed PCNs (24, 32, 33).*”

- Figure 1D caption: It would be helpful to mention in the caption that the second plasmid also contains wild-type repA under tight autoinhibition.

The caption now includes this helpful detail.

- Figure 2A: The lightened areas of the figure are difficult to see and should be slightly darker.

The figure has been updated accordingly for increased clarity.

- Figure 2A caption: Cuminic acid is misspelled.

The spelling is now corrected.

- Figure S3: The gates used for each round of screening should be included on the histograms.

The gates used for each successive round of screening are now indicated in Supplementary Fig. 3b as magenta overlays together with the histograms, along with the corresponding explanation in the caption.

- Figure S4: What was the metric used when selecting the ten variants?

We have selected the 10 colonies considering both their growth rate and their response to Cuminic acid indicating inducible PCN behaviour (now explicitly stated both in the section titled “Implementing inducible PCN control on a single plasmid” and in Supplementary Section 1.2.2). We have also updated Supplementary Fig. 3 to include data about each colony considering these two properties using a scatter plot.

- Cell growth is reported as normalized growth rate. The non-normalized growth rates should be included in the Supplement, namely when comparing TULIP in the different host strains.

All growth rate values have been updated to non-normalized “Specific Growth Rate (hr^{-1}).”

- Line 191: The proteins should be written as DnaA, DnaB, and DnaG.

References to these proteins have been corrected (the corresponding paragraph has been moved to Supplementary Section 1.3 where we also discuss how TULIP performs in different growth media).

- OriC is used in this manuscript to denote the pSC101 origin of replication. Use of this label can be somewhat confusing as OriC tends to be specifically used for the chromosomal origin of replication.

We thank the Reviewer for pointing out this issue. To address it, the labels have been corrected in all figures, and the region of DNA where replication initiates (white circles on the plasmid maps) are now indicated as “Ori” since pSC101 would refer to the whole origin of replication (including the regulatory proteins as well).

- Lines 282 and 290: The modeling section should be SI Section S3.

The reference has been corrected.

- Supplementary Section S3: The parameter values used should be justified with either references and/or a brief discussion within this section.

We thank the Reviewer for pointing out this shortcoming. To further strengthen the connection between theory and experiments both regarding the sensor modules in Figure 6, the toggle switch in Figure 7, and the NOT gate in Supplementary Fig. 14, we now include the justification for the parameter values alongside with the references in Supplementary Section 3.3.

References

1. H. Steel, R. Habgood, C. L. Kelly, A. Papachristodoulou, *PLOS Biology* **18**, 1 (2020).
2. J.-B. Lugagne, *et al.*, *Nature Communications* **8**, 1671 (2017).
3. J. Ratelade, *et al.*, *Applied and environmental microbiology* **75**, 3803 (2009).
4. J. B. Andersen, *et al.*, *Applied and environmental microbiology* **64**, 2240 (1998).
5. T. S. Gardner, C. R. Cantor, J. J. Collins, *Nature* **403**, 339 (2000).
6. J. W. Lee, *et al.*, *Molecular Cell* **63**, 329 (2016).

Reviewers' Comments:

Reviewer #1:

Remarks to the Author:

The authors have addressed all my concerns raised against the original manuscript. The results in the revised manuscript are now much more convincing following the authors decided to include several new features such as by incorporating the complex toggle switch circuit and CRISPRi-based regulatory/control layer, which has led to enhanced performance and increased applicability and versatility of the TULIP tool.

Baojun Wang

Reviewer #2:

Remarks to the Author:

Overall, the revisions and clarifications markedly improved the manuscript which now showcases the potential of inducible plasmid copy number control as a means for tuning gene expression and genetic circuits in standard laboratory *E. coli* strains. The authors have thoroughly expanded upon the characterization of TULIP with additional experiments addressing dynamics, plasmid stability, and burden. Notably, TULIP has little to no impact on cell fitness and is quite robust to metabolic burden. Furthermore, the authors use TULIP for circuit prototyping and also demonstrate that it can be used with multilayer control.

All my concerns have now been addressed, and I am happy to recommend this paper for publication in Nature Communications.

Minor Comments

- Main Text line 223, Supplementary Fig. 14 caption, SI Line 199: BW25113 is incorrectly written.

Response to Referees

We thank the Editor and the Reviewers for their constructive feedback that enabled us to improve our work and significantly expand the scope of the manuscript throughout the review process. Here we detail how we addressed the final comments that we have received.

Reviewer #1

The authors have addressed all my concerns raised against the original manuscript. The results in the revised manuscript are now much more convincing following the authors decided to include several new features such as by incorporating the complex toggle switch circuit and CRISPRi-based regulatory/control layer, which has led to enhanced performance and increased applicability and versatility of the TULIP tool.

Baojun Wang

We thank the Reviewer for their constructive feedback and suggestions that helped us not only improve the results presented in the original submission, but also sparked novel lines of research inquiry significantly expanding the scope of our work throughout review process.

Reviewer #2

Overall, the revisions and clarifications markedly improved the manuscript which now showcases the potential of inducible plasmid copy number control as a means for tuning gene expression and genetic circuits in standard laboratory *E. coli* strains. The authors have thoroughly expanded upon the characterization of TULIP with additional experiments addressing dynamics, plasmid stability, and burden. Notably, TULIP has little to no impact on cell fitness and is quite robust to metabolic burden. Furthermore, the authors use TULIP for circuit prototyping and also demonstrate that it can be used with multilayer control.

All my concerns have now been addressed, and I am happy to recommend this paper for publication in Nature Communications.

We thank the Reviewer for their constructive feedback and suggestions that helped us not only improve the results presented in the original submission, but also sparked novel lines of research inquiry significantly expanding the scope of our work throughout review process.

Minor Comments:

- Main Text line 223, Supplementary Fig. 14 caption, SI Line 199: BW25113 is incorrectly written. The incorrect spelling of BW25113 has been corrected throughout both the manuscript and the Supplementary Information.